# Genetic Insights into Congenital Cardiac Septal Defects—A Narrative Review

**DOI:** 10.3390/biology13110911

**Published:** 2024-11-07

**Authors:** Jorge L. Cervantes-Salazar, Nonanzit Pérez-Hernández, Juan Calderón-Colmenero, José Manuel Rodríguez-Pérez, María G. González-Pacheco, Clara Villamil-Castañeda, Angel A. Rosas-Tlaque, Diego B. Ortega-Zhindón

**Affiliations:** 1Department of Pediatric Cardiac Surgery and Congenital Heart Disease, Instituto Nacional de Cardiología Ignacio Chávez, Mexico City 14080, Mexico; jorgeluis.cervantes@gmail.com (J.L.C.-S.); angeltlaque@gmail.com (A.A.R.-T.); 2Department of Molecular Biology, Instituto Nacional de Cardiología Ignacio Chávez, Mexico City 14080, Mexico; unicanona@yahoo.com.mx (N.P.-H.); josemanuel_rodriguezperez@yahoo.com.mx (J.M.R.-P.); clara.vi.ca1@gmail.com (C.V.-C.); 3Department of Pediatric Cardiology, Instituto Nacional de Cardiología Ignacio Chávez, Mexico City 14080, Mexico; juanecalderon@yahoo.com.mx; 4Facultad de Ciencias de la Salud, Universidad Anáhuac México Norte, Mexico City 52786, Mexico; mgonzalezpacheco11@gmail.com; 5Programa de Doctorado en Ciencias Biomédicas, Universidad Nacional Autónoma de México, Mexico City 04510, Mexico; 6Dirección General de Calidad y Educación en Salud, Secretaría de Salud, Mexico City 06600, Mexico; 7Programa de Maestría y Doctorado en Ciencias Médicas, Odontológicas y de la Salud, Universidad Nacional Autónoma de México, Mexico City 04510, Mexico

**Keywords:** congenital heart disease, septal defects, cardiogenesis, genetics

## Abstract

During the embryonic development of vertebrates, the heart is the first organ to develop through cardiogenesis, a specialized and complex process that include several morphogenetic pathways. Any disruption that influences the onset expression of key role genes could impact the correct development of the heart and, consequently, the presence of congenital heart diseases (CHDs). These CHDs are defined as structural and conduction abnormalities and are considered to have multifactorial etiology. Within these malformations, the septal defects such as ventricular (VSD) and atrial septal defects (ASD), are the most common forms of CHDs. Studies have reported that CHDs are the result of genetic and environmental factors. Here, we review and summarize the role of genetics involved in cardiogenesis and congenital cardiac septal defects.

## 1. Introduction

Congenital heart diseases (CHDs) are abnormalities in the anatomy and function of the heart diagnosed in newborns, resulting from abnormal development of the fetal heart [1]; these anomalies account for 1% of live births globally [2]. The European Society of Cardiology (ESC) as well as The American Heart Association (AHA)/American College of Cardiology (ACC) guidelines categorize CHDs into simple, moderately complex, and complex, based on anatomical and physiological criteria. Nevertheless, the application of this standardized classification system encounters challenges in low- and middle-income countries due to inherent limitations within their healthcare systems [3]. As a result of substantial medical progress, survival rates have dramatically improved; nowadays, more than 90% of people with CHDs reach adulthood, and adults account for over two-thirds of the current population with CHDs in Western countries. CHDs are the most common congenital defect in humans, affecting approximately 1% of live newborns in the United States of America and Europe, with a survival rate of over 90% into adulthood. In the USA, approximately 81% of newborns with heart defects are expected to live to adulthood [4].

CHDs can be classified according to the age of onset, the affected cardiac chamber, or the pathophysiological mechanism [5]. The most common and long-used classification, cyanotic and no-cyanotic, is an excellent and convenient categorization due to its practicality and ease of understanding. For instance, cyanotic CHDs are characterized by the presence of cyanosis and commonly encompass the Tetralogy of Fallot (ToF), transposition of the great arteries (TGA), and septal defects with a right-to-left shunt. Conversely, no-cyanotic CHDs, which do not typically present cyanosis, include the coarctation of the aorta and septal defects with a left-to-right shunt, among others. Within these malformations, the septal defects, ventricular (VSDs) and atrial septal defects (ASDs), are the most common forms of CHDs [6].

Septal defects encompass a range of CHDs characterized by abnormal communication between the left and right heart chambers at atrial, ventricular, or both levels, resulting in a shunt between systemic and pulmonary circulation. When they occur in isolation, they are classified as simple CHDs; when they occur with associated abnormalities or a greater shunt, they are classified as moderately complex. Septal defects constitute the most prevalent subtype of CHDs on a global scale. VSDs and ASDs exhibit estimated incidences of 0.99 and 0.90 per 1000 live births, respectively. Collectively, the septal defects represent between 24.5 and 30.3% of all unrepaired CHDs, which underlines their significant contribution to the overall burden of this conditions [2,7]. The etiology of these defects is largely unknown, due to the lack of understanding of molecular and morphological processes involved in cardiac morphogenesis, as well as the limited access of genetic testing in some regions [8,9]. This review article highlights the relationship and synergy among genetic alterations, anatomical malformations that trigger physiological malfunctions, and the resulting symptoms in these pathologies. These relationships are relevant in current surgical, clinical, and molecular contexts due to the considerable incidence of CHDs previously mentioned.

The early detection of genetic alterations associated with septal defects is crucial for a timely and appropriate therapeutic approach, aiming for maximum benefits to the patient. Here, we review and summarize the role of genetics involved in cardiogenesis and congenital cardiac septal defects. Moreover, treatment regarding these congenital cardiac septal defects is also addressed.

## 2. Atrial Septal Defects

ASDs include any condition where there is a communication between the atrial cavities that allow the shunting of blood; these pathologies account for the most common CHDs diagnosed in adulthood (Figure 1) [10].

### 2.1. Pathophysiology of ASDs

The direction and degree of blood shunting in ASDs are mainly determined by the dimensions of the defect, the pressure gradient between atriums, and the difference in compliance between the right and left ventricles [11]. Patients are usually asymptomatic up until they reach their second to fourth decades of life, when the gradually increased pulmonary blood flow may lead to pulmonary vascular remodeling, appearing as clinical manifestations such as dyspnea on exertion, palpitations, and soft systolic *crescendo-decrescendo* murmurs [12].

The majority of small defects (<10 mm) are not associated with the enlargement of the right heart structures; contrarily, large long-standing shunts can result in myocardial stretch and injury [13]. ASDs are mainly divided into the ostium secundum type (which involves 80% of the cases), the ostium primum type (in 10% of the cases), and ASDs with sinus venosus or coronary sinus defect, which are the least frequent type. The ostium primum type can be associated with a common AV valve (atrioventricular septal defects (AVSD) [10].

The ASDs can represent an important cause of morbidity and mortality when they are not repaired before the patients reach 30 years, especially when it is accompanied by other congenital malformations or concomitant electrical disturbances. In patients with unrepaired ASD, overall mortality increases in adulthood because it can coexist with other acquired heart disease, resulting in complications such as stroke and ischemic heart disease, in addition to complications characteristic of this entity, such as pulmonary arterial hypertension (PAH) [10,11].

### 2.2. Treatment for ASDs

Although some ASDs close spontaneously, the surgical closure of the defect should not be postponed in the following cases: when there is hemodynamic instability or it is compromised by left-to-right shunting, when the right heart structures are dilated, when there is PAH due to volume overload, when growth retardation in children is observed, or when paradoxical embolism is observed [14]. Currently, a therapeutic option used for the correction of this heart disease is closure by percutaneous transcatheter intervention; however, in cases associated with other cardiac malformations or where the anatomy of the septal defect is not favorable for this type of closure, surgery is considered the ideal treatment. The surgical approach offers good long-term results and a low mortality rate (close to 0%), when considering conventional cardiac surgery up to the new minimally invasive surgical strategies, such as mini-sternotomy, subaxillary incisions, and mini-thoracotomy [15,16].

## 3. Ventricular Septal Defects

The failure of ventricular septation during embryonic development results in an opening between the right and left cavities that causes shunting during systole (Figure 2).

### 3.1. Pathophysiology of VSDs

A nomenclature has been well established to classify VSDs based on their size and location. Perimembranous defects can extend into the inlet, trabecular, or outlet septum, and they are the most frequent type (80%) of VSDs. Muscular types can differ in location and are occasionally multiple; they account for 15 to 20% of the defects. Outlet VSDs only represent 5% of all VSDs; they are located beneath the semilunar valves in the conal or outlet septum. Finally, there is the inlet or AVSD, which are positioned immediately inferior to the AV valve apparatus and are associated with a common AV valve [17]. VSDs may also occur with various degrees of malalignment of the septum, which also involve complex anomalies such as ToF, TGA, and congenitally corrected TGA (ccTGA) [18]. The size of the defects correlates with the hemodynamic impact, as large or multiple openings lead to complications and ventricular chamber dilation in early childhood. Additionally, the magnitude of shunting is dependent upon the vascular resistance within the pulmonary and systemic circulation. The clinical presentation of VSDs also varies according to the characteristics of the defects; small defects typically cause a turbulent flow that leads to a loud pansystolic murmur often associated with a palpable thrill, while in large defects, no murmur is audible. Patients with high pulmonary vascular resistance and shunt reversal show central cyanosis and a clubbing of fingers [19].

### 3.2. Treatment for VSDs

The spontaneous closure of small VSDs is common during childhood; nevertheless, the surgical intervention might be necessary if an infant shows inadequate development or heart failure symptoms that require medication. Patients eligible for closure at adult age are rare, since most of them have an insignificant shunt or have already developed pulmonary vascular disease.

Traditionally, individuals with large muscular VSDs underwent surgical closure via sternotomy with cardiopulmonary bypass; however, recent minimally invasive techniques for repairing VSDs have reduced recovery time and hospital stays. In addition, the closure of ventricular septal defects with devices through cardiac catheterization has allowed both pediatric and adult patients to receive treatment with this therapeutic option in recent years, instead of undergoing surgery. Although survival at 30-year follow-up of patients undergoing surgery is about 90%, patients with unrepaired defects have a higher lifetime mortality than the general population as a consequence of the development of VSD-related complications [20].

## 4. Cardiogenesis

During vertebrate embryonic development, the heart is the first organ to be formed, starting at around 15 days of development. At this stage, the heart progenitor cells specialize and form the cardiac crescent, expressing crucial genes such as *NKX2.5* and *GATA4*, which are necessary for heart development. The activation of these genes depends on signals from the underlying endoderm, including cerberus, BMP, and FGF-8 (Figure 3).

At 3 weeks of development, the cardiac cells migrate towards the midline to form a cardiac tube, composed of layers of endothelial and myocardial tissue. By the fourth week of development, the primitive heart undergoes a torsion that positions the atria above the ventricles and initiates the emergence of the aortic arch arteries.

The process of cardiac septation begins around the fourth developmental week when the primitive heart undergoes a torsion that positions the atria dorso-cephalically to the ventricles (the definitive right–left distribution of the cardiac cavities) and allows the emergence of the aortic arch arteries. Several essential genes, such as *NKX2.5*, *SRF*, *GATA4*, *TBX5*, and *HAND1* are involved in the differentiation of progenitor cell populations into cardiac cell lineages to originate the first heart field (FHF) which forms the left ventricle and the atrioventricular canal, and the second heart field (SHF), which contributes to the atria, right ventricle, and the outflow tract [29,30] (Figure 4).

During weeks 5 and 6 of development, the heart undergoes further division into four chambers, and the outflow tract separates into the pulmonary artery and the aorta, establishing pulmonary and systemic circulations. Initially, the formation of the primitive cardiac septum is facilitated by three main components: the septum primum, the endocardial cushions, and the primitive interventricular septum [38]. The septum primum, under the regulatory influence of *TBX5*, *GATA4*, and *NOS3* genes [39], grows apically and posteriorly from a muscular ridge, partially dividing the cavity into right and left atria, forming the ostium primum. Subsequently, the anterior and posterior endocardial cushions are fused, accompanied by the dorsal mesenchymal protrusion differentiation into myocardium, being downregulated by the *NKX2.5* gene. Concurrently, apoptosis regulated by the *ACTC1* gene [40] is induced in the cranial portion of the septum primum, forming fenestrations that evolve into the ostium secundum. The septation of the heart continues with the development of the septum secundum around the 33rd day of gestation, arising from the right atrium overlapping the ostium secundum, to form a right-to-left unidirectional valve called foramen ovale, which will eventually close after birth.

*TBX2*, *TBX3*, and *BMP2* genes, the nuclear factor of activated T-cells, and the cytoplasmic 2/3/4 (NFATC2/3/4) [41], in conjunction with the NOTCH, WNT, and transforming growth factor-β (TGF-β) signaling, induce the epithelial-to-mesenchymal transformation of endocardial cells, facilitating endocardial cushion’s formation that will contribute to the interventricular septum and the proximal portion of the ventricular outflow tracts [31,42].

The muscular component of the primitive interventricular septum originates in the apical direction from the primary fold that partially separates the ventricles, to align with the anterior and posterior endocardial cushions and the atrioventricular canal influenced by the *TBX6* gene, involved in the patterning of the cardiac myosin light chain [43,44], the *MYH6* gene that encodes the alpha myosin heavy chain [45], and the *TBX20* and *TBX5* genes that contribute to cardiomyocyte cell proliferation and lineage maturation [46]. Later on, the fusion of the atrioventricular endocardial cushions and the growth of the conal cushions of the outlet septum, under the regulatory influence of *TBX1*, finally form the membranous portion of the interventricular septum [47].

Morphological and anatomical studies have played a key role in the diagnosis and treatment of CHDs, while molecular biology has provided a better understanding of the underlying mechanisms; for example, different genetic pathways can influence the formation of an anatomical structure and the same congenital cardiac defects can be related to different mechanisms or genes. In this sense, Monroy-Muñoz et al. divided the genes and mechanisms involved in congenital heart diseases into four main categories: septation and atrioventricular connection defects, outflow tract or aortic arch defects, obstructive defects of the pulmonary artery and aorta, and right–left symmetry anomalies, such as the heterotaxy syndrome [1].

## 5. Genetic Factors

Several studies have extensively reported that genomic variations play a substantial role in the development of septal defects and aberrant cardiac morphology. It has been established that the gain or loss of function of morphogenetic transcription and growth factors affect embryonic cell populations, leading to deficiencies in alignment and the fusion of the septal components necessary to ensure full septation of the heart [48].

As the majority of septal defects do not have an identified etiology, the characterization of the genetic basis of septal defects is crucial; gene-to-gene interactions, gene-to-environment interactions, polygenic inheritance, or epigenetic mechanisms could be involved [49]. Recent advancements in molecular technologies enable the identification of gene variants in patients with septal defects, which facilitates the acquisition of clinical information that would help to determine patient management, thereby advancing the principles of precision medicine.

### 5.1. Chromosomal Abnormalities

Traditionally, CHD can be etiologically categorized as syndromic, non-syndromic inherited, or non-syndromic isolated. However, as the identification of genetic causes for CHDs is continuously progressing, the clear distinction between syndromic and non-syndromic etiologies becomes increasingly blurred, particularly due to the complexity of the subtle and variable behavior of phenotypic syndromic manifestations [50].

Although most septal defects occur sporadically, chromosomal abnormalities represent a significant cause of septal defects. These anomalies involving the loss or gain of entire chromosomes lead to widespread gene dysregulation, profoundly affecting genes critical for cardiac development. Consequently, 98% of fetuses with chromosomal abnormalities show at least one cardiac malformation [51]. Given these considerations, the chromosomal etiology of septal defects should be explored in any child presenting with a genetic syndrome and/or extracardiac anomalies. We herein describe some of the most prevalent chromosomal syndromes associated with septal defects.

For instance, DiGeorge syndrome is characterized by a 1.5 to 3 Mb microdeletion encompassing 30–40 genes associated with 22q11.2 deletion syndrome (22q11.2DS). A major contributor to CHD in this syndrome is the *TBX1* gene, which makes conotruncal defects such as VSD and ASD common [52].

In Down syndrome (trisomy 21), which is the most common aneuploidy, CHD occurs in about 45–50% of patients. The genetic basis for these defects involves several key genes on chromosome 21, such as *DSCAM*, *COL6A1*, *COL6A2*, *KCNJ6*, and *RCAN1*, which are crucial for the development of the atrioventricular septum and septum formation during fetal heart development [53,54]. A US population-based study revealed that septal defects were the most prevalent CHD in children with Down syndrome. Specifically, atrial ASDs were observed in 32.5%, VSDs in 20.6%, and atrioventricular septal defects in 17.4% of cases [51].

Patau syndrome (trisomy 13) and Edwards syndrome (trisomy 18) are also associated with a high prevalence of CHDs, observed in 18% and 31% of cases, respectively. In particular, VSDs have been reported in 77.4% of individuals with trisomy 18-related CHDs, while ASDs are observed in 45.2% of cases [55].

Holt–Oram syndrome is an autosomal dominant condition caused by mutations in the *TBX5* gene, which affect both limb and heart development. CHD occurs in approximately 91% of cases, with ASD being the most common defect (62%). Other associated heart problems include VSD, dilated cardiomyopathy (DCM), and arrhythmias, often requiring pacemaker implantation [56]. Genetic studies suggest that the effects of *TBX5* mutations can be influenced by interactions with other genes, such as *SALL4*. Additionally, microRNAs like miR-98-5p and miR-182-5p target *TBX5*, reducing its expression and potentially exacerbating heart development abnormalities and arrhythmias [57].

Mowat–Wilson syndrome, caused by mutations in the *ZEB2* gene, presents with unique facial features, intellectual disability, epilepsy, and various congenital anomalies, including CHD in 61% of patients. The syndrome affects the *SMAD1* and *GATA4* signaling pathways, which are crucial for septal formation [58]. The recent literature indicates that septal defects account for over 25% of cardiac abnormalities associated with Mowat–Wilson syndrome, alongside other simpler defects such as patent ductus arteriosus [59].

Noonan syndrome (NS) is a genetic disorder caused by mutations in several genes, most commonly *PTPN11*, *RAF1*, *SOS1*, and *BRAF*. Other genes, such as *RIT1*, *SOS2*, and *LZTR1*, have also been identified as contributors. CHD is prevalent in NS, with diverse phenotypic manifestations depending on the affected gene. Mutations in *PTPN11* are often associated with pulmonary valve stenosis and ASD, whereas *RAF1* mutations are linked to hypertrophic cardiomyopathy and ASD. Mutations in *SOS1* frequently result in pulmonary valve stenosis, while *BRAF* mutations can cause both *HCM* and valve abnormalities. *RIT1*, *SOS2*, and *LZTR1* have been associated with unique cardiac phenotypes, including valve malformations and dilated cardiomyopathy [60].

### 5.2. T-Box Family

The T-box proteins are a conserved family of transcription factors defined by a DNA-binding T-box domain of 180–200 amino acids, which allows them to regulate gene expression in a sequence-specific manner [48]. These proteins can act as either activators or repressors; some of them actually have a dual functionality depending on the cellular context. The family of T-box genes is present in various vertebrate species and has demonstrated a high conservation through evolution.

The phylogenetic relationships among these genes are evident in the sequence similarity within the region encoding this domain; as a consequence, they have been cataloged into five subfamilies, namely Brachyury (T), T-brain (Tbr1), TBX1, TBX2, and TBX6 [49], each one playing a distinct role in embryonic development. They influence cell fate specification, somite development, and the formation of the heart and limbs. Specific T-box genes, such as the *TBX1*, *TBX10*, *TBX15*, and *TBX22*, are vital for craniofacial development; while TBR1 and Eomes are key to brain development [61].

The T-box genes are essential regulators throughout the intricate process of cardiogenesis. Their influence encompasses the specification of cardiac mesoderm, the regionalization of the primitive heart tube, the formation of valves and septa, the recruitment of second heart field (SHF) cells to the outflow tract (OFT), and the establishment of cardiac conduction system. Notably, certain T-box genes indirectly impact the directional patterning of cardiac looping through their modulation of the left–right body axis. Eomes are critical for the initial specification of cardiac mesoderm, whereas the *TBX5* and *TBX20* genes are crucial for the chamber formation.

Mutations in these genes often result in severe developmental defects. The *TBX5* gene works with cofactors such as the *NKX2.5* and *GATA4* to activate chamber myocardial genes essential for chamber development. The *TBX20* gene also contributes to chamber myocardium development and endocardial cushion formation, impacting both embryonic and adult heart functions [62,63].

#### 5.2.1. TBX5

The *TBX5* gene is located on chromosome 12q24.21. It is expressed in endocardial cells destined to form the septum primum, where it plays a cell-autonomous role in cell survival, regulating endocardial growth and cardiomyocyte differentiation through interaction with *GATA4* and *NOS3* genes [39], which is vital for regulating cell survival and septum formation. As mentioned previously, it is well established that a reduction in the dosage of *TBX5* impairs heart development and heterozygous mutations in the *TBX5* gene causing Holt–Oram syndrome and resulting in septal defects. Additionally, the crucial cell-autonomous role in atrial septal formation implies that mutations in the *TBX5* gene might be present in patients with ASDs [64].

#### 5.2.2. TBX20

The *TBX20* gene is located on 7p14.2 and it encodes a transcription factor that belongs to the T-box family, crucial in cardiac development. The relationship between the *TBX20* gene and VSDs is based on the critical role in heart formation and development during embryogenesis. Studies in animal and human models have shown that mutations or alterations in the expression of *TBX20* can lead to defects in the formation of the interventricular septum, resulting in VSDs. The *TBX20* gene also regulates the expression of other genes involved in septal development and cardiac cell proliferation and differentiation [46].

#### 5.2.3. TBX2

The *TBX2* gene maps to 17q23.2 and suppresses the cardiac chamber gene program, directing cell differentiation and playing a critical role in the development of the atrioventricular canal myocyte through NOTCH and WNT signaling. Consequently, TBX2-expressing cells ultimately form the right ventricle and interventricular septum. Studies have identified *TBX2* gene promoter variants that disrupt microRNA binding, leading to altered gene expression and an increased risk of VSDs, as well as other outflow tract anomalies [41].

#### 5.2.4. TBX1

The *TBX1* gene is located on 22q11.21. It plays a pivotal role in cardiovascular morphogenesis via contributing to cardiac tube elongation and regulating the proliferation of mesenchymal precursor cells essential for the myocardialization and septation of the developing outflow tract. Evidence indicates that variations in the *TBX1* gene transcript levels are associated with cardiac and extracardiac developmental abnormalities. Haploinsufficiency of this gene has been well-established as the underlying cause of 22q11 deletion syndrome. Furthermore, studies have identified *TBX1* gene mutations linked to an increased susceptibility to non-syndromic ToF, VSDs, and other conotruncal anomalies [47].

#### 5.2.5. TBX6

Recent investigations have elucidated the critical role of the *TBX6* gene in the differentiation of pluripotent stem cells into mesoderm. Mapped to 16p11.2, the *TBX6* gene temporally regulates the inhibition of cardiovascular lineage specification, and significantly influences the patterning of cardiac muscle myosin light chain types [43]. Copy number variations (CNVs) and mutations within the *TBX6* gene have been linked to functional loss, resulting in the development of septal defects, other cardiac abnormalities, and congenital scoliosis [44].

### 5.3. MYH6

The *MYH6* gene is located on the 14q11.2 region and codes for the alpha myosin heavy chain (α-myosin), an important protein in cardiac muscle. This protein is essential for the contraction of myocytes, the muscle cells of the heart. Although MYH6 is most directly associated with cardiomyopathies and other conditions that affect the contractility of the heart, some research has suggested that mutations in the *MYH6* gene may also be involved in structural abnormalities of the heart, including the formation of the interventricular septum. These mutations can alter the proliferation, differentiation, and function of cardiac myocytes, which could indirectly contribute to defects in the development of the interventricular septum and, therefore, to VSDs [65].

### 5.4. CITED2

The *CITED2* (Cbp/p300-interacting transactivator with Glu/Asp-rich carboxy-terminal domain 2) maps to 6q24.1, and it encodes a transcriptional coactivator that interacts with several transcription factors and plays a crucial role in the development of the heart. Studies in animal models have shown that the deletion or dysfunction of the *CITED2* gene can result in congenital heart defects, including VSDs. CITED2 influences the activity of important signaling pathways and transcription factors such as HIF-1α and TFAP2, which are crucial for a proper development of the interventricular septum. The loss or alteration of the CITED2 function can lead to abnormal development of the heart, including the incomplete or defective formation of the interventricular septum, resulting in VSDs [66].

### 5.5. NKX2.5

NKX2.5 (transcription factor NKX2.5) is recognized as one of the most crucial factors for cardiac development. The gene is located on chromosome 5q35.1; it functions as a homeobox transcription factor expressed in progenitor cells of the first heart fields and the pharyngeal endoderm. NKX2.5 plays a pivotal role in tissue differentiation and in determining the temporal and spatial patterns during nearly all stages of heart development [67].

The functions of the *NKX2.5* gene depend heavily on its quality, quantity, and distribution, all of which can be influenced by single nucleotide variants. These variants can disrupt the downstream targets of its RNA transcript, potentially leading to congenital heart defects. The *NKX2.5* gene is known to be hypermutable, with variants found in up to 8% of patients with ASDs who have a familial history of the condition. In contrast, the overall detection frequency of the *NKX2.5* variants in sporadic cases of ASDs is between 1 and 4%, indicating that sporadic forms of the condition likely result from a diverse array of genetic factors. Moreover, mutations in the *NKX2.5* gene have been estimated to account for approximately 4% of ToF cases. This further underscores the significant role of this gene in cardiac development and its potential impact on congenital heart diseases [68].

In summary, the NKX2.5 is a vital homeobox transcription factor essential for heart development, influencing tissue differentiation and developmental patterns. Its function is contingent upon its genetic integrity, with mutations and variants linked to congenital heart conditions such as ASDs and ToF. Understanding the genetic variations and regulatory mechanisms of the *NKX2.5* gene is crucial for elucidating the genetic basis of these CHDs.

### 5.6. GATA4

The *GATA4* is a gene that codes for transcription factor (GATA binding protein 4) located on chromosome 8p23.1-p22; it plays a critical role in cardiac development and function, being expressed by all cardiac cells. The primary significance of GATA4 lies in its ability to integrate various cardiac transcription factors, cofactors, epigenetic regulators, and microRNAs. This integration is essential for several key processes in heart development, including the formation of the heart tube, the generation of the proepicardium, the separation and development of the outflow tract, myocardial angiogenesis, and the development of the cardiac conduction systemic [69].

The *GATA4* gene expression is tightly regulated by multiple factors, including the NKX2.5, the F-actin-binding protein NEXN, HYDIN, BMP signaling pathways, and others. This regulation is crucial because a decrease in GATA4 levels can disrupt cardiac gene regulatory networks, leading to an increased risk of abnormal development of the atrial and ventricular septa. Such disruptions can result in CHDs, which highlights the importance of precise *GATA4* regulation.

Furthermore, various mutations and variants in the *GATA4* gene have been identified in cases of sporadic ASDs. These mutations include both non-synonymous (affecting amino acid sequences) and synonymous changes, as well as noncoding and intronic variants. These genetic alterations can impact the function of *GATA4*, further emphasizing its critical role in maintaining normal cardiac development and function [70,71].

In summary, GATA4 is a pivotal transcription factor in cardiac biology, coordinating numerous elements necessary for heart development and function. Its expression and regulation are vital for normal cardiac structure and activity, with disruptions linked to congenital heart defects such as ASDs. Understanding the mechanisms governing the *GATA4* expression and function can provide valuable insights into cardiac development and the genetic basis of heart diseases.

### 5.7. ACTC1

Previously reported genetic markers associated with ASDs include Alpha-cardiac actin (*ACTC1*) located on chromosome 15q14, which is the only cardiac actin in the embryonic myocardium [40], crucial for cardiac contraction; additionally, *ACTC1* was the first causative gene described for autosomal dominant heart failure, as it can cause cell growth and polymerization defects, as well as induce apoptosis at the atrial and ventricular level [72]. Reports have demonstrated that downregulation in ACTC1 levels may lead to cardiac septal defects and cardiomyopathies [73]. Dominant *ACTC1* gene mutations have been identified in isolated ASDs (mainly the ostium secundum type), especially the mutation discovered in the 3′UTR that regulates translation, stability, and localization [74].

### 5.8. NOS3

The *NOS3* gene maps to chromosome 7q36, which codes to nitric oxide synthase 3. The nitric oxide produced by cardiomyocytes regulates myocardial relaxation and contractility and enhances coronary perfusion through angiogenesis. Additionally, eNOS has been shown to play a role in cardiomyogenesis by downregulating apoptosis and completing atrioventricular septation. Its activity is influenced by various conditions and external stimuli such as growth factors, histamine, adrenergic agonists, stress, and diabetes, many of which have been associated with higher risks of developmental defects [75]. Also, it has been described that polymorphisms of the *NOS3* are associated with an increased risk of ASDs [39].

### 5.9. NOTCH1

NOTCH1 is a critical regulator of cardiac septal formation, influencing epithelial-to-mesenchymal transition during heart development. It modulates pathways involved in cardiomyocyte proliferation (BMP10) and endocardial differentiation (EFNB2). NOTCH1 signaling, particularly in the endocardium and myocardium, is essential for trabecular myocardium formation and ventricular chamber compaction. Disruptions in NOTCH1 signaling can result in septation defects, leading to congenital heart conditions like hypoplastic left heart syndrome and ventricular wall and septal defects [76]. MIB1, NOTCH1, and GATA6 act synergistically in the developmental process of valves and the interventricular septum, while NOTCH1 and GATA6 insufficiency have been demonstrated to originate membranous VSDs [77].

### 5.10. MYBPC3

MYBPC3 is a crucial protein that plays a vital role in the regulation of cardiac muscle contraction and relaxation. It is part of the thick filament of the sarcomere, interacting with myosin and other proteins to modulate the contractile process. Mutations in the *MYBPC3* gene can disrupt the normal structure and function of the myosin-binding protein C3, leading to impaired cardiac muscle contraction and relaxation [78]. These mutations can be inherited in an autosomal dominant pattern, and they can vary in their effects, ranging from mild to severe forms of CHD. The molecular mechanisms by which *MYBPC3* mutations lead to CHD involve alterations in the protein’s ability to interact with other components of the sarcomere, such as myosin, troponin, and tropomyosin [79]. These interactions are critical for the proper functioning of the cardiac muscle, and their disruption can result in abnormal heart development and function, and therefore could lead to the development of novel therapeutic strategies [80].

### 5.11. TLL1

The *TLL1* (Tolloid-like 1) gene, located on chromosome 4q25, encodes a metalloprotease that processes procollagen C-propeptides, which takes a pivotal role in the development of the heart. Studies in animal models have shown that the deletion or dysfunction of the *TLL1* gene can result in congenital heart defects, and the homozygous silencing of this gene leads to death at mid-gestation due to defects in the blood circulatory system related to the lack of its proper flow because of the VSD and ASD, which were accompanied by dysplasia of the mitral valve [81].

As TLL1 is involved in the processing of extracellular matrix components, the loss or alteration of *TLL1* function can lead to the abnormal development of the heart. Moreover, the leading signaling path in which TLL1 may be involved in heart development seems to include BMP2 and BMP4, which are crucial for the atrioventricular septation and left–right patterning of the heart [82].

### 5.12. PITX2

The *PITX2* gene plays a crucial role in the development of the outflow tract and atrioventricular septum. PITX2 orchestrates the specification and migration of cardiac progenitor cells, facilitating the epithelial-to-mesenchymal transition required for septal formation. Furthermore, it regulates signaling pathways such as *BMP* and *FGF*, which are essential for cushion remodeling and myocardial patterning, and mutations are frequently associated with significantly diminished transcriptional activity. PITX2 abnormalities exhibit various septal defects, including unseptated atria, malformed endocardial cushions, and truncated outflow tracts, hindering normal cardiac development. Also, it was identified as a causative gene for the human Axenfeld–Rieger’s syndrome which also classically presents with concomitant CHD [83,84].

### 5.13. IRX4

The homeobox transcription factor IRX4 is a pivotal early marker of ventricular myocardium differentiation. In both the FHF and SHF of the cardiac crescent, IRX4 expression promotes the specification of left ventricular cardiomyocytes, upregulating HAND1 and downregulating ANF and alpha skeletal actin [28]. Under the regulatory influence of NKX2.5, IRX4 generates ventricular progenitor cells with the potential to differentiate into endothelial cells, smooth muscle cells, and cardiomyocytes [85]. Mutations in *IRX4* can disrupt the expression of genes essential for interventricular septum formation [86]. Table 1 shows the genes and genetic variants involved in congenital septal defects.

## 6. Discussion

In developmental complex diseases such as CHDs, the heterogeneous etiology is not fully understood. The indispensable role of the T-box gene family in cardiac development is underscored, as these transcription factors orchestrate a complex interplay of cellular processes from the earliest stages of mesoderm specification to the intricate formation of cardiac structures. Therefore, the pleiotropic effects of T-box genes, emphasizing their involvement in both intracardiac and extracardiac development are of great importance. On the other hand, specific genes such as *TBX5* and *TBX20* are directly linked to septal defects and a wide spectrum of genetic factors contributing to these conditions. Another group of genes such as *MYH6*, *CITED2*, and *NKX2.5* emphasizes the complex interplay between multiple genetic loci in the development of CHDs.

Additionally, the collaborative roles of *TBX5* with *NKX2.5* and *GATA4* genes in the chamber development highlight the synergistic effects of multiple genes. Although we focused on the in-depth analysis of reports from the specialized literature regarding the genetic association of congenital septal defects, the scarce knowledge of the pathogenic mechanisms of these diseases represents the main limitation of this type of approach.

This could be due to the large number of genes and susceptibility loci for CHDs, as well as the insufficient exploration of gene−gene interaction, interactions between genetic factors and environmental factors. Environmental factors may contribute to an estimated 10% of CHDs, and these factors include pre-gestational diabetes, comorbidities, in-utero exposure to alcohol, maternal infection, several medications, and so forth. Environmental exposures are relevant factors in the paradigm of the prognosis for CHDs [117,118,119].

Some genome-wide association studies (GWAS) have been performed in CHDs in different populations. A GWAS performed by Cordell et al. described a region on chromosome 4p16 close to the *MSX1* and *STX18* genes with common SNPs (rs6824295, rs16835979, and rs870142) that were associated with a risk of ASD [120].

Moreover, Hu et al. reported another GWAS in the Han Chinese population. This study included cases with ASD, VSD, and combined defects ASD/VSD. The authors found important associations on chromosome 1p12 (rs2474937 near *TBX15*), and chromosome 4q31.1 (rs1531070 in *MAML3*) [121]. Lahm et al. identified multiple risk loci through a GWAS in European patients with CHDs (4034 patients and 8486 controls). This study involved patients with various diagnoses, including septal defects, and the relevant genes were *ASIC2* and *STX18-AS1* [122].

On the other hand, the use of other technologies such as massive sequencing has emerged as a powerful tool for investigating the genetic variants contributing to CHDs. In this regard, Zhang et al. demonstrated that five gene variants (*NOTCH2*, *ATIC*, *MRI1*, *SLC6A13*, *ATP13A2*) were associated with VSD in the Chinese Tibetan population through whole exome sequencing [123]. Then, Zhang et al. identified six new pathogenic genes (*ACVRL1*, *ATP13A2*, *ERAP1*, *MRI1*, *TRAP1*, and *FBN2*) for Tibetan Familial with VSD using whole exome sequencing [124].

Another study by Grueso Cerón et al. characterized genomic variants associated with CHDs, identifying several genes including *NKX2.5*, *TBX20*, *GATA4*, *NOTCH1*, and *PTPN11*, which revealed 17% of intronic variants and 4.8% of missense variants in the southwestern Colombian population [125].

Conversely, environmental risk factors could also modify genetic risk in genetically predisposed individuals [126]. For instance, the analysis of epigenetic modifications such as DNA methylation and histone modifications can influence the formation and maturation of the heart. On numerous occasions, changes in DNA methylation caused increased variability on the methylated allele, likely related to a failure in maintenance or an adaptive response [127]. In particular, the imprinted gene showing a silencing pattern based on parental origin could play an important role in the disease. In addition, the environmental influences on imprinting centers include exposure to chemical pollutants and nutritional status in utero [128], both highly important during the first two weeks of heart development.

This epigenetic imprint can have a significant impact on heart development; therefore, it is a promising avenue to explore and relate epigenetic mechanisms to the inheritance aspects of congenital heart diseases. Future research should involve the application of technological advancements and large data to conduct comprehensive multi-omics analyses of CHDs, including genomics, epigenetics, transcriptomics, metabolomics, and proteomics. Recent studies utilizing omics technology have identified numerous differential proteins, metabolites, and key pathways in patients with VSDs who further develop pulmonary arterial hypertension [129]. In addition, molecular technologies such as candidate gene-based association studies, family-based studies, linkage disequilibrium analysis, whole genome or exome analyses, and mendelian randomization studies have been extensively used for researching complex cardiovascular diseases; these studies have described the interaction between modifier genes (epistasis) and non-genetic modifiers (epigenetic regulation) that influence the pathologies mentioned [130]. These technological advancements have revolutionized the paradigm of the management and prognosis of cardiovascular diseases, enabling the implementation of more personalized strategies.

Identifying novel genes and elucidating the precise mechanisms by which these genes interact with each other and with environmental factors will be essential to improve our understanding of CHDs and the molecular pathways that contribute to the pathophysiological process. Despite the significant advancements that have uncovered mechanisms involved in heart development, further research is necessary to elucidate the downstream targets of these processes and the intricate network of interactions with environmental factors, for a better molecular–phenotype correlation, and to potentially improve the diagnosis, prognosis, and treatment of septal defects.

## 7. Limitations

The present review contains a narrative descriptive with scientific knowledge concerning the genetic insights into congenital cardiac septal defects to support our viewpoint. However, it is important to mention that the search criteria may reflect some bias.

## 8. Conclusions

In recent years, the elucidation of the etiology of CHDs has massively improved due to the use of molecular technologies and the processing of data obtained through efficient bioinformatics tools. Although nowadays 90% of individuals with CHDs reach adulthood, the management of these conditions remains intricate, particularly in low- and middle-income countries where healthcare resources are limited. Cardiac septal defects (including ASD and VSD) are among the most common CHDs, impacting 1% of live births globally. The insights into the genetic basis of CHDs highlight cardiogenic genes which are critical for normal cardiac development, influencing processes such as septation and chamber formation. Thus, mutations in these genes can result in CHDs. Looking ahead, the integration of large scale international and advanced data will further clarify the genetic landscape of CHDs, and facilitate personalized treatment approaches, ultimately improving patient care and outcomes.

## Figures and Tables

**Figure 1 biology-13-00911-f001:**
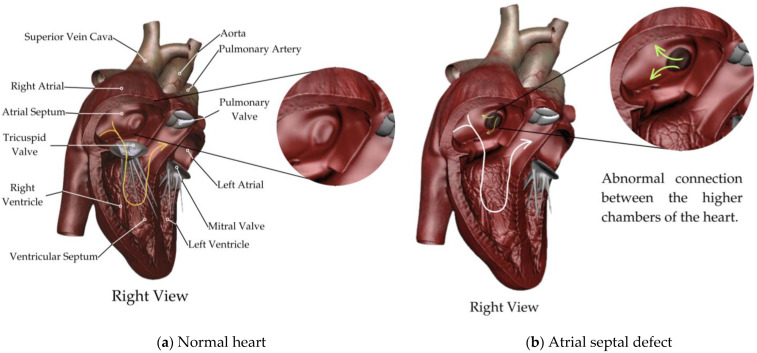
Atrial septal defect. A schematic representation of (**a**) normal heart; (**b**) atrial septal defect.

**Figure 2 biology-13-00911-f002:**
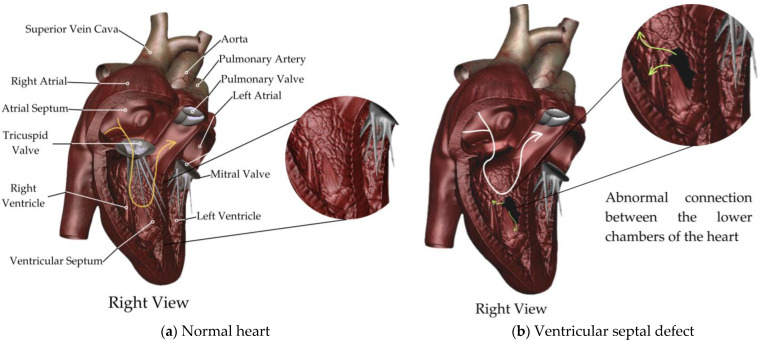
Ventricular septal defect. A schematic representation of (**a**) normal heart; (**b**) ventricular septal defect.

**Figure 3 biology-13-00911-f003:**
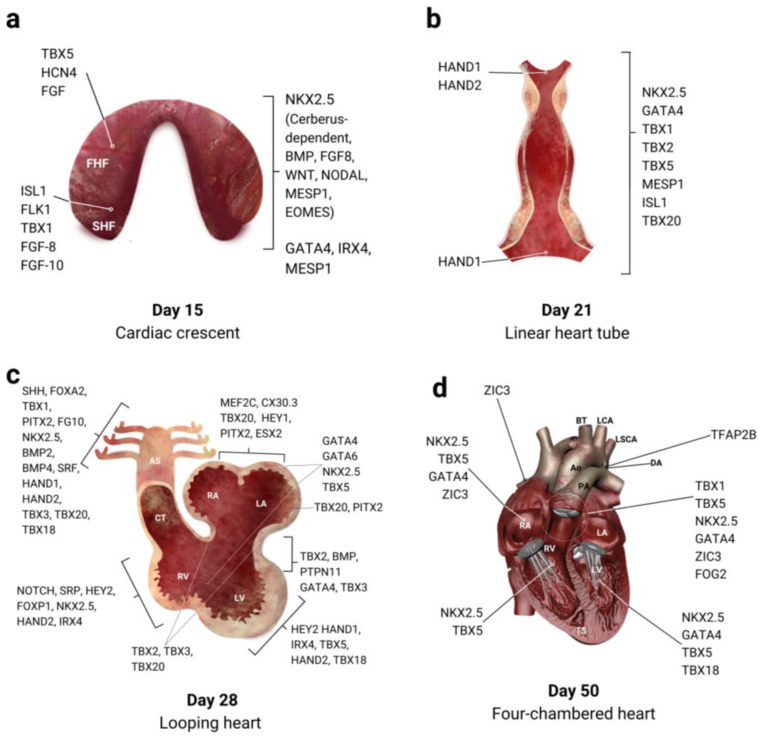
Schematic representation of the stages of cardiac morphogenesis. (**a**) Crescent stage at 15 days of development. (**b**) Linear tube stage, formed by the fusion of mesodermal cells at 21 days of development. (**c**) Stage of torsion and rotation of the heart tube and emergence of the aortic arch arteries from the outflow tract, at 28 days of development. (**d**) Stage of remodeling and growth of the ventricles with subsequent maturation of the heart and division of the circulation into systemic and pulmonary, from 50 days of development until birth. Ao: aorta; AS: aortic sac; BT: brachiocephalic trunk; CT: conotruncal ridges; DA: ductus arteriosus; LA: left atrium; LCA: left carotid artery; LSCA: left subclavian artery; LV: left ventricle; PA: pulmonary artery; RA: right atrium; RSCA: right subclavian artery; RV: right ventricle [21,22,23,24,25,26,27,28].

**Figure 4 biology-13-00911-f004:**
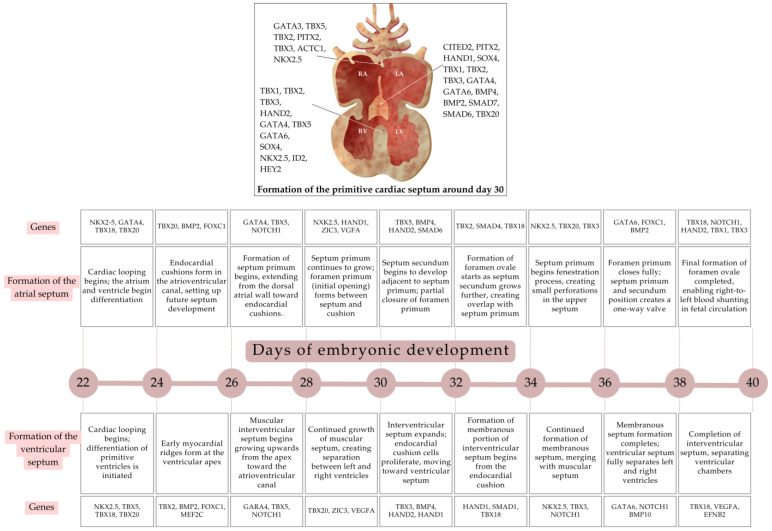
Formation of the primitive septum. A schematic representation of the formation of primitive septum between “Looping Heart” and the “Four-chambered Heart”. The embryonic age in days is shown from 22 to 40. The main events for the formation of the septum atrial (top) and ventricular (bottom) septum are noted in each horizon, highlighting some genes participating in each stage. LA: left atrium; LV: left ventricle; RA: right atrium; RV: right ventricle [22,23,24,31,32,33,34,35,36,37].

**Table 1 biology-13-00911-t001:** Genetic variants identified in ASD and VSD.

Gene	SNP/CNV	Variant *	Clinical Relevance and Condition **	Role in ASD/VSD
*ACTC1*	rs121912677	NM_002471.4(MYH6):c.2459T>A (p.Ile820Asn)	Pathogenic: ASD5	Mutations or reduced levels of *ACTC1* can lead to ASDs without associated cardiomyopathy. Specifically, the M123V substitution in ACTC1 reduces the protein’s affinity for myosin but retains some motor properties. A sporadic 17-bp deletion in *ACTC1* has been linked to CHDs. Other genetic variations can cause hypertrophic cardiomyopathy, particularly affecting the left ventricle, and are related to ASDs and VSDs [87].
	rs387906585	NM_005159.5(ACTC1):c.215_231del (p.Pro72fs)	Pathogenic: ASD5	
*NKX2.5*	rs1554093433	NM_004387.4(NKX2-5):c.711C>A (p.Tyr237Ter)	Pathogenic: ASD7, noncompaction cardiomyopathy, primary dilated cardiomyopathy, ventricular fibrillation.	Mutations in *NKX2.5* frequently lead to secundum ASDs and atrial ventricular conduction abnormalities, with notable mutations including T178M, Q170X, and Q198X. These mutations impact *NKX2.5* functionality differently, resulting in varying clinical outcomes. Variability in functional impairment is observed, where mutations leading to premature protein termination (e.g., Q170X) often result in more severe defects when compared with those affecting DNA binding alone (e.g., P59A) [79,88,89,90,91].
	rs104893900	NM_004387.4(NKX2-5):c.533C>T (p.Thr178Met)	Pathogenic: ASD7.	
	rs104893901	NM_004387.4(NKX2-5):c.508C>T (p.Gln170Ter)	Pathogenic: ASD7.	
	rs104893903	NM_004387.4(NKX2-5):c.592C>T (p.Gln198Ter)	Pathogenic: ASD7.	
	rs606231359	NM_004387.4(NKX2-5):c.228_229del (p.Pro77fs)	Pathogenic: ASD7.	
	rs137852683	NM_004387.4(NKX2-5):c.896A>G (p.Asp299Gly)	Pathogenic: ASD7, AVSD somatic.	
	rs606231360	NM_004387.4(NKX2-5):c.262del (p.Ala88fs)	Pathogenic: ASD7.	
	rs104893906	NM_004387.4(NKX2-5):c.568C>T (p.Arg190Cys)	Pathogenic: ASD7.	
	rs104893907	NM_004387.4(NKX2-5):c.768T>G (p.Tyr256Ter) NM_004387.4(NKX2-5):c.768T>A (p.Tyr256Ter)	Pathogenic: ASD7.	
	rs137852686	NM_004387.4(NKX2-5):c.547A>G (p.Lys183Glu)	Pathogenic: AVSD somatic.	
	rs387906773	NM_004387.4(NKX2-5):c.44A>T (p.Lys15Ile)	Pathogenic: ASD7.	
	rs387906774	NM_004387.4(NKX2-5):c.380C>T (p.Ala127Val) NM_004387.4(NKX2-5):c.380C>A (p.Ala127Glu)	Uncertain significance: NKX2.5 disorder.	
		NM_004387.4(NKX2-5):c.380C>A (p.Ala127Glu)	Pathogenic: ASD7.	
	rs375086983	NM_004387.4(NKX2-5):c.848C>A (p.Pro283Gln)	Pathogenic: VSD3. Uncertain significance: ASD7, conotruncal heary malformations, hypoplastic left heart syndrome, TOF, congenital hypotiroidism, cardiovascular phenotype.	
	rs387906775	NM_004387.4(NKX2-5):c.175C>G (p.Pro59Ala)	Pathogenic: VSD3.	
	rs387906776	NM_004387.4(NKX2-5):c.769C>A (p.Pro257Thr)c	Uncertain significance: ASD7, conotruncal heart malformations, hypoplastic left heart syndrome, ToF, congenital hypotiroidism, cardiovascular phenotype.	
		NM_004387.4(NKX2-5):c.769C>G (p.Pro257Ala)	Pathogenic: VSD3.	
*GATA4*	rs104894073	NM_001308093.3(GATA4):c.889G>C (p.Gly297Arg)	Pathogenic: VSD1.	Various mutations in the *GATA4* gene have been linked to ASDs and VSDs. This variation has been G-to-A transition, E359del 839C-T transition (thr280-to-met), and a 928A-G transition (met310-to-val) in families with ASDs and VSDs showing reduced activation of the ANP promoter [92,93,94,95,96,97].
		NM_001308093.3(GATA4):c.889G>T (p.Gly297Cys)	Pathogenic: ASD2.	
		NM_001308093.3(GATA4):c.889G>A (p.Gly297Ser)	Pathogenic: VSD2, TGA.	
	rs1585703301	NM_001308093.3(GATA4):c.1078del (p.Glu360fs)	Pathogenic: ASD2.	
	rs104894074	NM_001308093.3(GATA4):c.155C>T (p.Ser52Phe)	Pathogenic: ASD2.	
	rs56298569	NM_001308093.3(GATA4):c.949C>G (p.Gln317Glu)	Pathogenic: ASD2, AVSD4.	
	rs368489876	NM_001308093.3(GATA4):c.1078G>T (p.Glu360Ter)	Pathogenic: AVSD4.	
		NM_001308093.3(GATA4):c.1078G>A (p.Glu360Lys)	Pathogenic: VSD1. Uncertain significance: AVSD4.	
		NM_001308093.3(GATA4):c.889G>A (p.Gly297Ser)	Pathogenic: ASD2, TGA.	
	rs104894073	NM_001308093.3(GATA4):c.889G>C (p.Gly297Arg)	Pathogenic: VSD1.	
		NM_001308093.3(GATA4):c.889G>T (p.Gly297Cys)	Pathogenic: ASD2.	
		NM_001308093.3(GATA4):c.889G>A (p.Gly297Ser)	Pathogenic: ASD2, TGA.	
		NM_001308093.3(GATA4):c.127C>T (p.Arg43Trp)	Pathogenic: VSD1. Uncertain significance: Cardiovascular phenotype.	
	rs387906771	NM_001308093.3(GATA4):c.842C>T (p.Thr281Met)	Pathogenic: ASD2. Uncertain significance: ASD4 [98,99].	
	rs387906772	NM_001308093.3(GATA4):c.931A>C (p.Met311Leu)	Uncertain significance: ASD4.	
		NM_001308093.3(GATA4):c.931A>T (p.Met311Leu)	Uncertain significance: ASD4, VSD1, GATA4 related disorder.	
		NM_001308093.3(GATA4):c.931A>G (p.Met311Val)	Pathogenic: ASD2 Uncertain significance: GATA4 related disorder [100].	
	rs804280	NM_001308093.3(GATA4):c.1000+56C>A	Pathogenic: CHD, Benign—ASD4, GATA4 related disorder	
	rs115099192	NM_001308093.3(GATA4):c.1223C>A (p.Pro408Gln)	Pathogenic: TOF, VSD1. Uncertain significance: ASD2, AVSD4, VSD1 Benign: AVSD4 Likely benign: Testicular anomalies with or without CHD.	
		NM_001308093.3(GATA4):c.1223C>G (p.Pro408Arg)	Uncertain significance: ASD4.	
	rs1062219	NM_001308093.3(GATA4):c.*426C>T	Uncertain significance.	
	rs748737164	NM_001308093.3(GATA4):c.1312G>A (p.Gly438Arg)	Likely Pathogenic: ASD2. Uncertain significance: AVSD4.	
*GATA6*	rs387906816	NM_005257.6(GATA6):c.551G>A (p.Ser184Asn)	Pathogenic/Likely Pathogenic: ASD9. Pathogenic: ToF Benign/Likely benign: AVSD.	GATA6, a key cardiac transcription factor involved in heart development, was analyzed resulting in several variants causing loss of protein function, highlighting Hispanic and Chinese populations [101,102,103].
	rs587777710	NM_005257.6(GATA6):c.712G>A (p.Gly238Arg)	Uncertain significance.	
		NM_005257.6(GATA6):c.712G>T (p.Gly238Ter)	Pathogenic: Abnormal cardiovascular system morphology, congenital diaphragmatic hernia, pancreatic hypoplasia-diabetes, CHD.	
*TBX20*	rs137852954	NM_001077653.2(TBX20):c.456C>G (p.Ile152Met)	Uncertain significance: ASD4, Cardiovascular phenotype.	The most widely accepted hypothesis is that mutations in the *TBX20* gene can disrupt cardiac development through both gain-of-function and loss-of-function mechanisms. Research has shown that mutations such as Ile152-to-Met (I152M) and Gln195-to-Stop (Q195X) impair protein function by affecting DNA binding and truncating essential domains. The I121M mutation, associated with enhanced transcriptional activity, suggests a gain-of-function effect [104,105].
	rs137852955	NM_001077653.2(TBX20):c.583C>T (p.Gln195Ter)	Pathogenic: ASD4.	
	rs267607106	NM_001077653.2(TBX20):c.363C>G (p.Ile121Met)	Pathogenic: ASD4.	
*MYH6*	rs267606903	NM_002471.4(MYH6):c.2459T>A (p.Ile820Asn)	Pathogenic: ASD3.	The identification of a new locus for atrial septal defect on chromosome 14q12, linked to a missense mutation (I820N) in *MYH6* promoter, highlights the role of structural protein dysfunction in congenital heart disease. This mutation disrupts the interaction with its regulatory light chain, and underscores the connection between transcription factors like TBX5 and structural proteins in the development of heart malformations [65,106,107,108].
	rs368451573	NM_002471.4(MYH6):c.3604G>A (p.Val1202Met)	Uncertain significance: ASD, hypertrophic cardiomyopathy 1 and 14, ASD3, Dilated cardiomyopathy 1EE and dominant, sick sinus syndrome3, cardiovascular phenotype. Likely benign: hypertrophic cardiomyopathy 14.	
*TBX5*	rs1555223259	NM_181486.4(TBX5):c.1221C>G (p.Tyr407Ter)	Pathogenic: ASD optimum secundum, mitral regurgitation, VSD.	Pathogenic variants in *TBX5*, *TFAP2B*, and *PTPN11* were identified in 10% of families of CHDs, while likely disease-causal variants in additional CHDs candidate genes were found in 33% of families, indicating a broad genetic basis for CHDs [56,109,110].
*CITED2*	rs779637348	NM_006079.5(CITED2):c.510_536del (p.163GSSTPGGSG [1])	Pathogenic: VSD2.	These novel mutations in the *CITED2* gene, found in patients with various types of congenital heart defects, are likely to impair its function. Specifically, the deletions and insertions identified affect the protein’s ability to regulate transcription factors HIF1A and TFAP2C, which are critical for normal cardiac development. The presence of these mutations in patients with ASD and other cardiac anomalies underscores the *CITED2* potential role in congenital heart disease [111,112].
		6q23.3 (CITED2) 27-BP INS NT534 (p.Gly178_Ser179ins9)	Pathogenic: ASD8.	
	rs531316452	NM_006079.5(CITED2):c.581GCGGCA [2] (p.196SG [1])	Pathogenic: ASD8.	
*TLL1*	rs137852953	NM_012464.5(TLL1):c.1885A>G (p.Ile629Val)	Pathogenic: ASD6.	These genetic variants, found in various domains of *TLL1*, could contribute to ASDs by affecting protein function. Their presence in patients with ASDs are associated with complications like interatrial septum aneurysm highlighting their potential impact on cardiac morphogenesis [113,114].
	rs137852952	NM_012464.5(TLL1):c.713T>C (p.Val238Ala)	Pathogenic: ASD6.	
	rs137852951	NM_012464.5(TLL1):c.544A>C (p.Met182Leu)	Pathogenic: ASD6.	
		NM_012464.5(TLL1):c.544A>G (p.Met182Val)	Uncertain significance.	

* Nomenclature according to ClinVar. ** Nomenclature according to ACMG guidelines, OMIM [115,116].

## Data Availability

Not applicable.

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
