# Peer review of "Genetic Insights into Congenital Cardiac Septal Defects—A Narrative Review"

_biology, 2024, doi:10.3390/biology13110911_

Round 1

Reviewer 1 Report

Comments and Suggestions for Authors

The article by Cervantes-Salazar and colleagues provides a comprehensive overview of the genetic basis of congenital cardiac septal defects. While the topic is significant, the manuscript requires substantial revisions to improve both the clarity of the writing and the development of ideas. In some sections, specific claims would benefit from stronger supporting evidence. I encourage the authors to take the time to address these issues in detail, as doing so will significantly enhance the manuscript's overall impact. Below, I expand on several key areas that I consider essential.

Major issues:

First, the authors need to clarify and deepen the explanation of the main objective of the article. They state, “We summarize current evidence linking genetic factors to these structural heart abnormalities,” but the article does not clearly describe these factors. The manuscript combines multiple topics—genetic factors related to both syndromic and isolated defects—and mentions single nucleotide variants (SNVs) and copy number variations (CNVs). However, the clinical relevance of these genetic factors (e.g., variants of uncertain significance, likely pathogenic, pathogenic according to ACMG guidelines) is omitted. As a reader, I found it challenging to identify the article’s specific contributions and the inclusion of updated information. Furthermore, I recommend including data from articles related to massive sequencing and genome-wide association studies (GWAS) and expanding the genetic analysis using resources like OMIM or ClinVar, rather than relying solely on article databases.

The authors mentioned that the information was gathered using PubMed, Scopus, and Web of Science. However, as an up-to-date review, they should have conducted a systematic review of the genetic factors associated with ASD and VSD. A non-systematic search may introduce bias in the information presented. For example, in the article (10.1590/1678-4685-GMB-2020-0142), a pathogenic variant in the TBX5 gene is linked to familial congenital heart disease (CHD), including ASD and VSD. This reference, along with others involving genes like NKX2.5 and MYH6 (10.1093/ehjci/ehaa946.3724), as well as newly proposed ASD-related genes like STX18 (10.1038/ng.2637), were not included in the list of RS references.

The authors should consider specifying whether the described genes are associated with patent foramen ovale, ostium secundum defect, ostium primum defect, sinus venosus defect, or coronary sinus defect.

Minor issues:

In the cardiogenesis section several affirmations should be verified carefully; for example, “…the first heart field (FHF) cells forming the right ventricle and the atrioventricular canal, while the second heart field (SHF) gives rise to the atria, left ventricle, and much of the outflow tract”. However, SHF is linked to right ventricle development, while FHF is related to the left ventricle. I recommend consulting reviews like Transcriptional pathways in second heart field development (10.1016/j.semcdb.2007.01.001) and Unveiling Complexity and Multipotentiality of Early Heart Fields (10.1161/RES.0000000000000534). Furthermore, the information in this section is confusing and requires a complete reorganization. In spite, the authors should restructure it to identify the relevance of the genes discussed; for example, describing the septum primum as originating from a muscular ridge could highlight the importance of muscular differentiation factors in the etiology of anomalies in this septum.

The authors should verify if they are using adequately the terms developmental week and gestational weeks. Whereas the first term is related to fertilization, the second one is related to the Last Menstrual Period (LMP). I recommend looking over article 10.3390/jcdd9060187 to separate the concepts, and therefore, correct the weeks in the article.

Regarding Figure 3, the authors should restructure it to focus on genes relevant to septum primum, septum secundum, endocardial cushions, muscular ridge, and conal ridge development. Some typos were found, such as “cerebrus” instead of “cerberus” in 3a. Moreover, the authors mix human and mouse nomenclature, which should be corrected. Other corrections should be done in gene expression, for example, Tbx5 is more strongly associated with the left ventricle (Early patterning and specification of cardiac progenitors in gastrulating mesoderm 10.7554/eLife.03848), and its expression is primarily seen in the atria and left ventricle, while Tbx20 is expressed in the atria, right ventricle, outflow tract, and the atrioventricular canal (10.1074/jbc.M314041200).

Upon reviewing Table 1, several issues arise. The title refers to “ASV” instead of “VSD.” Focusing on just one of the described genes, NKX2.5, reveals multiple inaccuracies: rs3729753 and rs2277923 are classified as benign and therefore, not associated with septal defects, whereas rs34402828 is an SNV/SNP is related to HAND1, not NKX2.5. A thorough revision of this gene and others is necessary to avoid these issues. Additionally, the main table presents a list of reference SNPs (RS), but it is not informative in its current form. 

The authors should use the genetic nomenclature for SNV/SNP and their effect, for example in GATA4 (in the role in ASD VSD, the description is suboptimal, the authors described “G-to-A transition, E359del 839C-T transition (thr280-to-met) and a 928A-G transition (met310-to-val)” instead of the most actual description as, p.E359del, c.839C>T (p.Thr280Met), c.928A>G (p.Met310Val). Right now I’m pretty unsure of where the G to A transition described initially is located. This lack of consistency of nomenclature is observed throughout the text.

In the MYH6 section of the table, the authors affirm that mutations underscore the connection between transcription factors, despite this one of the references used to sustain this is related to the MYH6 promoter; therefore, the authors clarified and accurately described. Additionally, the MYH6 section (5.7) references FLNC instead of myosin heavy chain, so this needs correction.

Lastly, the inclusion of FABP4 is pointless. This protein has been described as a biomarker related to post-operative outcomes for ASD or VSD patients (10.1161/JAHA.121.024072) and is not relevant as the article is focused on etiology. In spite, the cited reference pertains to an aging population, not the defects being discussed. Similarly, TLL1 is mentioned only in the table without any explanation of its significance. Instead of these genes, the authors should consider including other ASD/VSD-related genes, such as PITX2, NOTCH1, and MYBPC3.

As outlined, several sections require significant revisions. Consequently, the article will need a considerable amount of time and work for further development before it can be reconsidered.

Comments on the Quality of English Language

In summary, the manuscript requires significant improvements in English grammar. Some sections use inappropriate verbs or lack sufficient connectors, which disrupts the flow of ideas. Strengthening these aspects will greatly enhance readability and comprehension.

Author Response

We thank the reviewer for the time invested in reviewing our manuscript. The observations made were very important to improve the revised version.

First, the authors need to clarify and deepen the explanation of the main objective of the article. They state, “We summarize current evidence linking genetic factors to these structural heart abnormalities,” but the article does not clearly describe these factors.

Replay:

Thank you for the suggestion and we agree with the reviewer with this important point.

In order to clarify this issue, we have homogenized the objective of the work in the sections of the simple summary (lines 29-39), abstract (lines 35-37), and in the last paragraph of the introduction (lines 84-86). The following sentences was added:

“Here, we review and summarize the role of genetics involved in cardiogenesis and congenital cardiac septal defects. Moreover, treatment regarding these congenital cardiac septal defects is also addressed”.

The clinical relevance of these genetic factors (e.g., variants of uncertain significance, likely pathogenic, pathogenic according to ACMG. Also, I recommend including data from articles related to massive sequencing and genome-wide association studies (GWAS) and expanding the genetic analysis using resources like OMIM or ClinVar, rather than relying solely on articles databases. 

Replay:

We are in accordance with the reviewer. Now, in table 1, we described in column 4 information of variants with their clinical relevance. Also, we expanded the genetic analysis using resources like OMIM and ClinVar. Please see, Table 1 (lines 542). Moreover, we included data from articles related to massive sequencing and genome-wide association studies (GWAS) (lines 567 -588).

“Some genome-wide association studies (GWAS) have been performed in CHDs in different populations. A GWAS performed by Cordell et al. described a region on chromosome 4p16 close to the MSX1 and STX18 genes with common SNPs (rs6824295, rs16835979, and rs870142) that were associated with risk of ASD [122].

Moreover, Hu et al., reported another GWAS in the Han Chinese population. This study included cases with ASD, VSD and combined defects ASD/VSD. The authors found important associations on chromosome 1p12 (rs2474937 near TBX15), and chromosome 4q31.1 (rs1531070 in MAML3) [123].

Lahm et al., identified multiple risk loci through a GWAS in patients European with CHDs (4034 patients and 8486 controls). This study involved patients with various diagnoses, including septal defects and the relevant genes were ASIC2 and STX18-AS1 [124].

On the other hand, the use of other technologies such as massive sequencing, has emerged as a powerful tool for investigating the genetic variants contributing to CHDs. In this regard, Zhang et al, demonstrated that five gene variants (NOTCH2, ATIC, MRI1, SLC6A13, ATP13A2) were associated with VSD in Chinese Tibetan population through whole exome sequencing [125]. Then, Zhang et al., identified six new pathogenic genes (ACVRL1ATP13A2ERAP1MRI1TRAP1, and FBN2) for Tibetan Familial with VSD using whole exome sequencing [126].

Another study by Grueso Cerón et al., characterized genomic variants associated with CHDs identified several genes including NKX2.5TBX20GATA4NOTCH1, and PTPN11, which revealed that 17% of intronic variants and 4.8% of missense variants in southwestern Colombian population [127]. “

The following references were included in the manuscript:

  1. Cordell, H.J.; Bentham, J.; Topf, A.; Zelenika, D.; Heath, S.; Mamasoula, C.; Cosgrove, C.; Blue, G.; Granados-Riveron, J.; Setchfield, K. Genome-wide association study of multiple congenital heart disease phenotypes identifies a susceptibility locus for atrial septal defect at chromosome 4p16.  Genet.201345, 822-824.
  2. Hu, Z.; Shi, Y.; Mo, X.; Xu, J.; Zhao, B.; Lin, Y.; Yang, S.; Xu, Z.; Dai, J.; Pan, S. A genome-wide association study identifies two risk loci for congenital heart malformations in Han Chinese populations.  Genet2013, 45, 818-821.
  3. Lahm, H.; Jia, M.; Dreßen, M.; Wirth, F.; Puluca, N.; Gilsbach, R.; Keavney, B.D.; Cleuziou, J.; Beck, N.; Bondareva, O. Congenital heart disease risk loci identified by genome-wide association study in European patients.  Clin. Invest.2021131, e141837.
  4. Zhang, X.; Zhen, D.; Li, X.; Yi, F.; Zhang, Z.; Yang, W.; Li, X.; Sheng, Y. ; Liu, X. ; Jin, T.; et al. NOTCH2, ATIC, MRI1, SLC6A13, ATP13A2 Genetic Variations are Associated with Ventricular Septal Defect in the Chinese Tibetan Population Through Whole-Exome Sequencing. Pharmgenomics Pers. Med202316, 389-400.
  5. Zhang, X.; Zhen, D. ; Yi, F. ; Zhang, T.; Li, X.; Wang, Y.; Li, X.; Sheng, Y.; Liu, X.; Jin, T, et al. Identification of Six Pathogenic Genes for Tibetan Familial Ventricular Septal Defect by Whole Exome Sequencing.  Surg. Res2024296:18-28.

 127. Grueso Cerón, A.L.; Arturo-Terranova, D.; Satizábal Soto, J.M. Characterization of genomic variants associated with congenital heart disease in patients from southwestern Colombian. Heliyon202310, e23678.

The authors mentioned that the information was gathered using Pubmed, Scopus, and Web of Science. However, as an up-to-date review, they should have conducted a systematic review of the genetic factors associated with ASD and VSD. A non-systematic search may introduce bias in the information presented.

Replay:

We appreciate the comment, and we understand the reviewer's concern. In this regard, we described the manuscript considering the following:

In the present manuscript (a narrative review), we synthesize information into a legible-friendly format with a global scope, on the genetics basis of congenital cardiac septal defects. As opposed to, systematic reviews a specific question through detailed and exhaustive bibliographic searches. In this sense, we currently, do not have quantitative methods to design and carry out this type of analysis to analyze and include all the evidence available in scientific databases.

In order to clarify this issue, we adapted the title of the manuscript “Genetic insights into congenital cardiac septal defects – A narrative review” (line 2).

Also, we added a limitations section with the following statement:

  1. Limitations (622-625):

 The present review contains a narrative descriptive with scientific knowledge concerning the genetic into congenital cardiac septal defects to support our viewpoint. However, it is important to mention that the search criteria may reflect some bias.

The authors should consider specifying whether the described genes are associated with patent foramen ovale, ostium secundum defect, ostium primum defect, Sinus venosus defect or coronary sinus defect.

 Replay:

We sincerely thank them for their comments. In the manuscript, we have mainly addressed in Figure 4 and Table 1 the interrelationship between genes and their function in the process of atrial and interventricular septal formation. Although his suggestion to specify genes according to their anatomical and physiological variants is valid, it is complicated; since, for example, patent foramen ovale is a concept that integrates both structural and physiological aspects, given that it originates from incomplete fusion of the septum primum with the septum secundum after birth. This phenomenon is influenced by changes in pressures due to the onset of pulmonary circulation and increased left atrial pressure. In this context, a pathophysiological explanation becomes more relevant than the identification of a specific gene. Therefore, through the previously mentioned materials (Figure and Table), a detailed explanation of the genes involved in the development of the atrial septum is offered; however, the expression with its specific anatomical localization encompasses physiological concepts that transcend the objective of this manuscript.

MINOR ISSUES

In the cardiogenesis section several affirmations should be verified carefully.

For example, “…the first heart field (FHF) cells forming the right ventricle and the atrioventricular canal, while the second heart field (SHF) gives rise to the atria, left ventricle, and much of the outflow tract”.

However, SHF is linked to right ventricle development, while FHF is related to the left ventricle. I recommend consulting reviews like Transcriptional pathways in second heart field development (10.1016/j.semcdb.2007.01.001) and Unveiling Complexity and Multipotentiality of Early Heart Fields (10.1161/RES.0000000000000534).

Replay:

We are grateful for the reviewer's guidance. Now, in this version we corrected this important observation. Also, both references suggested by the reviewer were added in the manuscript.

The following sentence was modified e included in the cardiogenesis section:

… the first heart field (FHF) which forms the left ventricle and the atrioventricular canal, and the second heart field (SHF) that contributes to the atria, right ventricle, and the outflow tract [29,30] (lines 193-195).

  1. Black, B.L. Transcriptional pathways in second heart field development. Semin. Cell Dev. Biol. 2007, 18, 67-76.
  2. Zhang Q, Carlin D, Zhu F, Cattaneo P, Ideker T, Evans SM, Bloomekatz J, Chi NC. Unveiling Complexity and Multipotentiality of Early Heart Fields. Res. 2021, 129, 474-487.

Furthermore, the information in this section is confusing and requires a complete reorganization. In spite, the authors should restructure it to identify the relevance of the genes discussed; for example, describing the septum primum as originating from a muscular ridge could highlight the importance of muscular differentiation factors in the etiology of anomalies in this septum.

Replay:

We appreciate the reviewer's valuable recommendation. We adapted the cardiogenesis section that includes which describes more precise information (Lines 167-240)

“During vertebrate embryonic development, the heart is the first organ to be formed, starting at around 15 days of development. At this stage, the heart progenitor cells specialize and form the cardiac crescent, expressing crucial genes such as NKX2.5 and GATA4, which are necessary for heart development. The activation of these genes depends on signals from the underlying endoderm, including cerberus, BMP, and FGF-8 (Figure 3).

At 3 weeks of development, the cardiac cells migrate towards the midline to form a cardiac tube, composed of layers of endothelial and myocardial tissue. By the fourth week of development, the primitive heart undergoes a torsion that positions the atria above the ventricles and initiates the emergence of the aortic arch arteries.

The process of cardiac septation begins around the fourth developmental week when the primitive heart undergoes a torsion that positions the atria dorso-cephalically to the ventricles (the definitive right-left distribution of the cardiac cavities) and allows the emergence of the aortic arch arteries. Several essential genes, such as NKX2.5, SRF, GATA4, TBX5, and HAND1 are involved in the differentiation of progenitor cell populations into cardiac cell lineages to originate the first heart field (FHF) which forms the left ventricle and the atrioventricular canal, and the second heart field (SHF) that contributes to the atria, right ventricle, and the outflow tract [29,30].

During weeks 5 and 6 of development, the heart undergoes further division into four chambers, and the outflow tract separates into the pulmonary artery and the aorta, establishing pulmonary and systemic circulations. Initially, the formation of the primitive cardiac septum is facilitated by three main components: the septum primum, the endocardial cushions, and the primitive interventricular septum [31]. The septum primum, under the regulatory influence of TBX5, GATA4, and NOS3 genes [32] grows apically and posteriorly from a muscular ridge, partially dividing the cavity into right and left atria, forming the ostium primum. Subsequently, the anterior and posterior endocardial cushions are fused, accompanied by the dorsal mesenchymal protrusion differentiation into myocardium, being downregulated by the NKX2.5 gene. Concurrently, apoptosis regulated by the ACTC1 gene [33] is induced in the cranial portion of the septum primum, forming fenestrations that evolve into the ostium secundum. The septation of the heart continues with the development of the septum secundum around the 33rd day of gestation, arising from the right atrium overlapping the ostium secundum, to form a right-to-left unidirectional valve called foramen ovale, that will eventually close after birth (Figure 4).

TBX2, TBX3, and BMP2 genes, the nuclear factor of activated T-cells, and the cytoplasmic 2/3/4 (NFATC2/3/4) [40]; in conjunction with the NOTCH, WNT and the transforming growth factor-β (TGF-β) signaling, induce epithelial-to-mesenchymal transformation of endocardial cells, facilitating endocardial cushion’s formation that will contribute to the interventricular septum and the proximal portion of the ventricular outflow tracts [41,42].

The muscular component of the primitive interventricular septum originates in the apical direction from the primary fold that partially separates the ventricles, to align with anterior and posterior endocardial cushions and the atrioventricular canal influenced by the TBX6 gene, involved in the patterning of cardiac myosin light chain [43,44], the MYH6 gene that encodes the alpha myosin heavy chain [45], TBX20 and TBX5 genes that contribute to cardiomyocyte cell proliferation and lineage maturation [46]. Later on, the fusion of the atrioventricular endocardial cushions and the growth of the conal cushions of the outlet septum, under the regulatory influence of TBX1, finally form the membranous portion of the interventricular septum [47].

Morphological and anatomical studies have played a key role in the diagnosis and treatment of CHDs, while molecular biology has provided a better understanding of the underlying mechanisms; for example, as different genetic pathways can influence the formation of an anatomical structure and after as the same congenital cardiac defect can be related to different mechanisms or genes. In this sense, Monroy-Muñoz et al. reported the genes and mechanisms involved in congenital heart diseases into four main categories: septation and atrioventricular connection defects, outflow tract or aortic arch defects, obstructive defects of the pulmonary artery and aorta, and right-left symmetry anomalies, such as the heterotaxy syndrome [1].”

  1. Monroy-Muñoz, I.E.; Pérez-Hernández, N.; Vargas-Alarcón G, Ortiz-San Juan G, Buendía-Hernández A, Calderón-Colmenero J, Ramírez-Marroquín S, Cervantes-Salazar JL, Curi-Curi P, Martínez-Rodríguez N.; et al. Changing the paradigm of congenital heart disease: from the anatomy to the molecular etiology. Med. Mex. 2013, 149, 212-219.
  2. Black, B.L. Transcriptional pathways in second heart field development. Semin. Cell Dev. Biol. 2007, 18, 67-76.
  3. Zhang Q, Carlin D, Zhu F, Cattaneo P, Ideker T, Evans SM, Bloomekatz J, Chi NC. Unveiling Complexity and Multipotentiality of Early Heart Fields. Res. 2021, 129, 474-487.
  4. Burns T, Yang Y, Hiriart E, Wessels A. The Dorsal Mesenchymal Protrusion and the Pathogenesis of Atrioventricular Septal Defects. Cardiovasc. Dev. Dis. 2016, 3, 29.
  5. Nadeau, M.; Georges, R.O.; Laforest, B.; Yamak, A.; Lefebvre, C.; Beauregard, J.; Paradis, P.; Bruneau, B.G.; Andelfinger, G.; Nemer, M. An endocardial pathway involving Tbx5, Gata4, and Nos3 required for atrial septum formation. Natl. Acad. Sci. U. S. A. 2010, 107, 19356-19361.
  6. Frank, D.; Yusuf Rangrez, A.; Friedrich, C.; Dittmann, S.; Stallmeyer, B.; Yadav, P.; Bernt, A.; Schulze-Bahr, E.; Borlepawar, A.; Zimmermann, W.H.; et al. Cardiac α-Actin (ACTC1) Gene Mutation Causes Atrial-Septal Defects Associated With Late-Onset Dilated Cardiomyopathy. Genom. Precis. Med. 2019, 12, e002491.
  7. Wang, J.; Zhang, R.R.; Cai, K.; Yang, Q.; Duan, W.Y.; Zhao, J.Y.; Gui, Y.H.; Wang, F. Susceptibility to congenital heart defects associated with a polymorphism in TBX2 3' untranslated region in the Han Chinese population. Res. 2019, 85, 378-383.
  8. Lin, C. J.; Lin, C. Y.; Chen, C. H.; Zhou, B.; Chang, C. P. Partitioning the heart: mechanisms of cardiac septation and valve development. Development. 2012, 139, 3277-3299.
  9. Ye, C.; Yang, C.; Zhang, H.; Gao, R.; Liao, Y.; Zhang, Y.; Jie, L. ; Zhang, Y. ; Cheng, T.; Wang, Y.; et al. Canonical Wnt signaling directs the generation of functional human PSC-derived atrioventricular canal cardiomyocytes in bioprinted cardiac tissues. Stem Cell. 2024, 31, 398-409.
  10. Yano, Y.; Iimura, N.; Kojima, N.; Uchiyama, H. Non-neural and cardiac differentiating properties of Tbx6-expressing mouse embryonic stem cells. Ther. 2016, 3, 1-6.
  11. Sadahiro, T.; Isomi, M.; Muraoka, N.; Kojima, H.; Haginiwa, S.; Kurotsu, S.; Tamura, F.; Tani, H.; Tohyama, S.; Fujita, J. Tbx6 Induces Nascent Mesoderm from Pluripotent Stem Cells and Temporally Controls Cardiac versus Somite Lineage Diversification. Stem Cell. 2018, 23, 382-395.
  12. Huang, S.; Wu, Y.; Chen, S.; Yang, Y.; Wang, Y.; Wang, H.; Li, P.; Zhuang, J.; Xia, Y. Novel insertion mutation (Arg1822_Glu1823dup) in MYH6 coiled-coil domain causing familial atrial septal defect. J. Med. Genet. 2021, 64, 104314.
  13. Chakraborty, S.; Yutzey, K.E. Tbx20 regulation of cardiac cell proliferation and lineage specialization during embryonic and fetal development in vivo. Biol. 2012, 363, 234-246.
  14. Zhang, M.; Li, F.X.; Liu, X.Y.; Hou, J.Y.; Ni, S.H, Wang, J.; Zhao, C.M.; Zhang, W.; Kong, Y.; Huang, R.T.; et al. TBX1 loss-of-function mutation contributes to congenital conotruncal defects. Ther. Med. 2018, 15, 447-453.

Regarding figure 3, the authors should restructure it to focus on genes relevant to septum primum, septum secundum, endocardial cushions, muscular ridge, and conal ridge development. Some typos were found, such as “cerebrus” instead of “cerberus” in 3a. Moreover, the authors mix human and mouse nomenclature, which should be corrected. Other corrections should be done in gene expression, for example, Tbx5 is more strongly associated with the left ventricle (Early patterning and specification of cardiac progenitors in gastrulating mesoderm 10.7554/eLife.03848), and its expression is primarily seen in the atria and left ventricle, while Tbx20 is expressed in the atria, right ventricle, outflow tract, and the atrioventricular canal (10.1074/jbc.M314041200).

Replay:

Thank you for the recommendations. The spelling errors have been corrected, including changing "cerebrus" to "cerberus." Additionally, the human and mouse nomenclature has been standardized throughout the text.

Regarding the development of the septum primum, septum secundum, endocardial cushions, muscular ridge, and conal ridge, this process extends around day 22 to 44-46 days of embryonic development. The point of view you raise is interesting to create a comprehensive structure that encompasses all aspects. However, your suggestions were considered, and Figure 4 (lines 211-216) was generated to provide more detail on embryonic development from day 22 to day 40. The figure illustrates the progression from heart looping to the formation of the heart chambers, capturing key stages in the septum development process, where the formation of the septum is detailed and extended separately; ventricular (bottom) and atrial (top). This approach aimed to capture the critical stages in the development of the septum as clearly as possible.

Upon reviewing Table 1, several issues arise. The title refers to “ASV” instead of “VSD.” Focusing on just one of the described genes, NKX2.5, reveals multiple inaccuracies: rs3729753 and rs2277923 are classified as benign and therefore, not associated with septal defects, whereas rs34402828 is an SNV/SNP is related to HAND1, not NKX2.5. A thorough revision of this gene and others is necessary to avoid these issues. Additionally, the main table presents a list of reference SNPs (RS), but it is not informative in its current form. 

Replay:

Thanks for this suggestion. We adequate the table 1 and we have deleted SNPs/CNVs classified as benign. Also, we reviewed carefully genetic variants described with the correct association for these septal defects. Finally, we improved Table 1 to provide a better understanding and relevance of the content. Please see line 542.

The authors should use the genetic nomenclature for SNV/SNP and their effect.

Replay:

We agree with the reviewer. Now, on table 1 we have modified the genetic nomenclature for these genetic variants with their effects as established by the nomenclature to ACMG guidelines, OMIM and ClinVar. Please see Table 1 on lines 542.

In the MYH6 section of the table, the authors affirm that mutations underscore the connection between transcription factors, despite this one of the references used to sustain this is related to the MYH6 promoter; therefore, the authors clarified and accurately described. Additionally, the MYH6 section (5.7) references FLNC instead of myosin heavy chain, so this needs correction.

Replay:

Thanks to the reviewer for this observation. Now in this point, we included the correct references and in the table 1, we added MYH6 promoter.

Zuo, J.Y.; Chen, H,X.; Liu, Z.G.; Yang, Q.; He, G.W. Identification and functional analysis of variants of MYH6 gene promoter in isolated ventricular septal defects. B.M.C. Med Genomics. 2022, 15, 213.

Lastly, the inclusion of FABP4 is pointless. This protein has been described as a biomarker related to post-operative outcomes for ASD or VSD patients (10.1161/JAHA.121.024072) and is not relevant as the article is focused on etiology. Similarly, TLL1 is mentioned only in the table without any explanation of its significance. Instead of these genes, the authors should consider including other ASD/VSD-related genes, such as PITX2, NOTCH1, and MYBPC3.

Replay:

We agree with the reviewer. In the revised version, we delete FABP4 and we added information of NOTCH1, MYBPC3, TLL1 and PITX2, according to the reviewer suggestion (lines 480-529).

“5.9. NOTCH1

NOTCH1 is a critical regulator of cardiac septal formation, influencing epithelial-to-mesenchymal transition during heart development. It modulates pathways involved in cardiomyocyte proliferation (BMP10) and endocardial differentiation (EFNB2). NOTCH1 signaling, particularly in the endocardium and myocardium, is essential for trabecular myocardium formation and ventricular chamber compaction. Disruptions in NOTCH1 signaling can result in septation defects, leading to congenital heart conditions like hypoplastic left heart syndrome and ventricular wall and septal defects [76]. MIB1, NOTCH and GATA6 act synergistically in de developmental process of valves and the interventricular septum, while NOTCH1 and GATA6 insufficiency have demonstrated to originate membranous VSDs [77].

5.10. MYBPC3

MYBPC3 is a crucial protein that plays a vital role in the regulation of cardiac muscle contraction and relaxation. It is part of the thick filament of the sarcomere, interacting with myosin and other proteins to modulate the contractile process. Mutations in the MYBPC3 gene can disrupt the normal structure and function of the myosin-binding protein C3, leading to impaired cardiac muscle contraction and relaxation [78]. These mutations can be inherited in an autosomal dominant pattern, and they can vary in their effects, ranging from mild to severe forms of CHD. The molecular mechanisms by which MYBPC3 mutations lead to CHD involve alterations in the protein's ability to interact with other components of the sarcomere, such as myosin, troponin, and tropomyosin [79]. These interactions are critical for the proper functioning of the cardiac muscle, and their disruption can result in abnormal heart development and function, and therefore could lead to the development of novel therapeutic strategies [80].

5.11. TLL1

The TLL1 (Tolloid-like 1) gene, located on chromosome 4q25, encodes a metalloprotease that processes procollagen C-propeptides, which takes a pivotal role in the development of the heart. Studies in animal models have shown that deletion or dysfunction of the TLL1 gene can result in congenital heart defects, the homozygous silencing of this gene lead to death at mid-gestation due to defects in the blood circulatory system related to the lack of its proper flow because of the VSD and ASD, which were accompanied by dysplasia of the mitral valve [81]. As TLL1 is involved in processing of extracellular matrix components, the loss or alteration of TLL1 function can lead to abnormal development of the heart. Moreover, the leading signaling path in which TLL1 may be involved in heart development seems to include BMP2 and BMP4, which are crucial for the atrioventricular septation and left-right patterning of the heart [82].

5.12. PITX2C

The PITX2 gene plays a crucial role in the development of outflow tract and atrioventricular septum. PITX2 orchestrates the specification and migration of cardiac progenitor cells, facilitating the epithelial-to-mesenchymal transition required for septal formation. Furthermore, it regulates signaling pathways such as BMP and FGF, which are essential for cushion remodeling and myocardial patterning, mutations are frequently associated with significantly diminished transcriptional activity. PITX2 abnormalities exhibit various septal defects, including unseptated atria, malformed endocardial cushions, and truncated outflow tracts, hindering normal cardiac development, also, it was identified as a causative gene for the human Axenfeld-Rieger's syndrome which also classically presents with concomitant CHD [83,84].”

  1. Ye, S.; Wang, C.; Xu, Z.; Lin, H.; Wan, X.; Yu, Y.; Adhicary, S.; Zhang, J. Z.; Zhou, Y.; Liu, C.; et al. Impaired Human Cardiac Cell Development due to NOTCH1 Deficiency. Circ. Res. 2023, 132, 187-204.
  2. Piñeiro-Sabarís, R.; MacGrogan, D.; De La Pompa, J.L. Intricate MIB1-NOTCH-GATA6 Interactions in Cardiac Valvular and Septal Development.  Cardiovasc. Dev. Dis. 2024, 11, 223.
  3. Alsters, S.; Wong, L.; Peferoen, L.; Niessen, H. W. M.; Bikker, H.; Elting, M. W.; Houweling, A. C. Fatal neonatal hypertrophic cardiomyopathy caused by compound heterozygous truncating MYBPC3 mutation.  Heart J. 2019, 27, 282-283.
  4. Perrot, A.; Rickert-Sperling, S. Human Genetics of Ventricular Septal Defect.  Exp. Med. Biol. 2024, 1441:505-534.
  5. Manhas, A.; Jahng, J.W.S.; Vera, C.D.; Shenoy, S.P.; Knowles, J.W.; Wu, J.C. Generation of two iPSC lines from hypertrophic cardiomyopathy patients carrying MYBPC3 and PRKAG2 variants. Stem Cell Res. 2022, 61:102774.
  6. Sieroń, A.L.; Stańczak, P. ASD--lessons on genetic background from transgenic mice with inactive gene encoding metalloprotease, Tolloid-like 1 (TLL1).  Sci Monit. 2006, 12, RA17-RA22.
  7. Sieron, L.; Lesiak, M.; Schisler, I.; Drzazga, Z.; Fertala, A.; Sieron, A.L. Functional and structural studies of tolloid-like 1 mutants associated with atrial-septal defect 6. Biosci Rep. 2019, 39,
  8. Ma, H.Y; Xu, J; Eng, D; Gross , M.K; Kioussi ,C. Pitx2-mediated cardiac outflow tract remodeling.  Dyn. 2013, 242, 456-468.
  9. Zhao, C. M.; Peng, L. Y.; Li, L.; Liu, X. Y.; Wang, J.; Zhang, X. L.; Yuan, F.; Li, R. G.; Qiu, X. B.; Yang, Y. Q. PITX2 Loss-of-Function Mutation Contributes to Congenital Endocardial Cushion Defect and Axenfeld-Rieger Syndrome. PLoS One 2015, 10, e0124409.

Reviewer 2 Report

Comments and Suggestions for Authors

The heart septation defect is one of the main categories of congenital heart disease (CHD). They can affect the septation of the atria leading to atrial septal defect (ASD) and septation of ventricles leading to ventricular septal defect (VSD). In this review, the authors attempt to show the current genetic evidence of septal defects during cardiogenesis through databases such as PubMed, Scopus, and Web of Science, thus providing potential evidence for the diagnosis, prognosis, and treatment of ventricular septal defects. Overall, this could be an interesting and informative review manuscript. However, there are a couple of issues that need to be improved upon by the author.

1.    In Figures 1 and 2, it is useful to show normal and abnormal hearts without and with atrial septal defects and to clearly label the major parts of the heart structure. For example, the left and right ventricles, the left and right atria, the septum, the interventricular septum, and the different valves.

2.    An illustration of the formation of the primitive cardiac septum is desirable.

3.    Line 337, “5.7. MYH6” is confused, since MYH6 is not belong to T-BOX family. It should be 5.1. T-Box family, then 5.1.1 TBX5 etc., then 5.2. MYH6 etc.

4.    Chromosomal abnormalities may also be associated with a high prevalence of congenital cardiac ventricular septal defects. The authors may shed more light on this point.

5.    Whereas IRX4 and PITX2C also play an important role in the development of the cardiac septum, their mutations have been associated with cardiac septal defects. The authors would ideally need to comment on these factors as well.

Comments on the Quality of English Language

na

Author Response

We thank the reviewer for the time invested in reviewing our manuscript. The observations made were very important to improve the revised version.

  1. In Figures 1 and 2, it is useful to show normal and abnormal hearts without and with atrial septal defects and to clearly label the major parts of the heart structure. For example, the left and right ventricles, the left and right atria, the septum, the interventricular septum, and the different valves.

Replay:

The suggestion to show normal and abnormal hearts, with and without atrial septal defects, and clearly label the major parts of the heart structure has been implemented. In response, parts (a) of Figures 1 and 2 were added to illustrate the normal heart structure compared to the pathological heart discussed. Lines 112-113 for the atrial septal defect and lines 149- 151 for the ventral septal defect.

(a) Normal heart

(b) Atrial septal defects

Figure 1. Atrial septal defect. A schematic representation of (a) normal heart; (b) atrial septal defect.

(a) Normal heart

(b) Ventricular septal defect

Figure 2. Ventricular septal defect. A schematic representation of (a) normal heart; (b) ventricular septal defect.

  1. An illustration of the formation of the primitive cardiac septum is desirable.

Replay:

Based on the recommendation and due to the complexity of septum formation, Figure 4 was added, where the formation of the septum is detailed and extended separately; ventricular (bottom) and atrial (top). However, due to the complexity of the formation of the primitive cardiac septum, which starts around day 22 and concludes by days 42-46 of embryonic development, the process is challenging to represent in a single figure. The formation of the ventricular and atrial septum differs in terms of expression and structural development, making it difficult to encompass the entire process in one illustration.

As a result, the figure was designed to highlight key points in time for the formation of the septum, as well as to provide a schematic time-space representation of the septum formation around day 30, attempting to capture an intermediate stage between the "Looping Heart" and the "Four-chambered Heart”. This approach aimed to capture the critical stages in the development of the septa as clearly as possible (Lines 210 -216).

References for Figure 4:

  1. Furtado, M.B.; Biben, C.; Shiratori, H.; Hamada, H.; Harvey, R.P. Characterization of Pitx2c expression in the mouse heart using a reporter transgene. Dyn. 2011, 240, 195-203.
  2. George, R.M.; Firulli, A.B. Hand Factors in Cardiac Anat. Rec. (Hoboken). 2019, 302, 101-107.
  3. Greulich, F.; Rudat, C.; Kispert A. Mechanisms of T-box gene function in the developing heart. Res. 2011, 91, 212-222.
  4. Lin, C.J.; Lin, C.Y.; Chen, C.H.; Zhou, B.; Chang, C.P. Partitioning the heart: mechanisms of cardiac septation and valve development. Development 2012, 139, 3277-3299.
  5. Nakajima, Y.; Yamagishi, T.; Hokari, S.; Nakamura, H. Mechanisms involved in valvuloseptal endocardial cushion formation in early cardiogenesis: roles of transforming growth factor (TGF)-beta and bone morphogenetic protein (BMP).  Rec. 2000, 258, 119-127.
  6. Webb, S.; Brown, N. A.; Anderson, R. H. Formation of the atrioventricular septal structures in the normal mouse. Res. 1998, 82, 645-656.
  7. Garg, V.; Muth, A. N.; Ransom, J. F.; Schluterman, M. K.; Barnes, R.; King, I. N.; Grossfeld, P. D.; Srivastava, D. Mutations in NOTCH1 cause aortic valve disease. Nature 2005, 437, 270-274.
  8. Cowan, C. A.; Yokoyama, N.; Saxena, A.; Chumley, M. J.; Silvany, R. E.; Baker, L. A.; Srivastava, D.; Henkemeyer, M. Ephrin-B2 reverse signaling is required for axon pathfinding and cardiac valve formation but not early vascular development.  Biol. 2004, 271, 263-271.
  9. Carlson B.M. Cardiovascular System. In Human Embryology and Developmental Biology, 6th ed.; Editorial services DRK; Publisher: Elsevier, Avda. Josep Tarradellas, 20-30, 1st Floor, 08029, Barcelona, Spain, 2019, 409–415.

  1. Line 337, “5.7. MYH6” is confused, since MYH6 is not belong to T-BOX family. It should be 5.1. T-Box family, then 5.1.1 TBX5 etc., then 5.2. MYH6 etc.

Replay:

The issue regarding the categorization of MYH6, which does not belong to the T-BOX family, was addressed. The section was reorganized to include the T-Box family under 5.2, followed by subtopics such as 5.2.1 TBX5, and MYH6 was moved to a separate section, 5.3, for clarity. Lines 336, 347, 357, 365, 376 for the T-Box family and 385 for MYH6.

  1. Chromosomal abnormalities may also be associated with a high prevalence of congenital cardiac ventricular septal defects. The authors may shed more light on this point.

Replay:

Your comment about the relationship between chromosomal abnormalities and the prevalence of congenital cardiac ventricular septal defects was considered. Additional information has been incorporated into “5. Genetic factors” as “5.1 Chromosomal abnormalities”. The following section was added (lines 255-307):

“ 5.1. Chromosomal abnormalities

Traditionally, CHD can be etiologically categorized as syndromic, non-syndromic inherited, or non-syndromic isolated. However, as the identification of genetic causes for CHDs is continuously progressing, the clear distinction between syndromic and non-syndromic etiologies becomes increasingly blurred, particularly because due to the complexity of the subtle and variable behavior of phenotypic syndromic manifestations [50].

Although most septal defects occur sporadically, chromosomal abnormalities represent a significant cause of septal defects. These anomalies involving the loss or gain of entire chromosomes, lead to widespread gene dysregulation, profoundly affecting genes critical for cardiac development. Consequently, 98% of fetuses with chromosomal abnormalities show at least one cardiac malformation [51]. Given these considerations, chromosomal etiology of septal defects should be explored in any child presenting with a genetic syndrome and/or extracardiac anomalies. We herein describe some of the most prevalent chromosomal syndromes associated with septal defects.

For instance, DiGeorge syndrome is characterized by a 1.5 to 3 Mb microdeletion encompassing 30–40 genes associated with 22q11.2 deletion syndrome (22q11.2DS). A major contributor to CHD in this syndrome is the TBX1 gene, which makes conotruncal defects such as VSD and ASD common [52].

In Down syndrome (trisomy 21), which is the most common aneuploidy, CHD occurs in about 45–50% of patients. The genetic basis for these defects involves several key genes on chromosome 21, such as DSCAM, COL6A1, COL6A2, KCNJ6, and RCAN1, which are crucial for the development of the atrioventricular septum and septum formation during fetal heart development [53,54]. A US population-based study revealed septal defects being the were the most prevalent CHD in children with Down syndrome. Specifically, atrial ASD were observed in 32.5%, VSD in 20.6%, and atrioventricular septal defects in 17.4% of cases [51].

Patau syndrome (trisomy 13) and Edwards syndrome (trisomy 18) are also associated with a high prevalence of CHD, observed in 18% and 31% of cases, respectively. In particular, VSDs have been reported in 77.4% of individuals with trisomy 18-related CHDs, while ASDs are observed in 45.2% of cases [55].

Holt-Oram syndrome is an autosomal dominant condition caused by mutations in the TBX5 gene, which affect both limb and heart development. CHD occurs in approximately 91% of cases, with ASD being the most common defect (62%). Other associated heart problems include VSD, dilated cardiomyopathy (DCM), and arrhythmias, often requiring pacemaker implantation [56]. Genetic studies suggest that the effects of TBX5 mutations can be influenced by interactions with other genes, such as SALL4. Additionally, microRNAs like miR-98-5p and miR-182-5p target TBX5, reducing its expression and potentially exacerbating heart development abnormalities and arrhythmias [57].

Mowat-Wilson syndrome, caused by mutations in the ZEB2 gene, presents with unique facial features, intellectual disability, epilepsy, and various congenital anomalies, including CHD in 61% of patients. The syndrome affects the SMAD1 and GATA4 signaling pathways, which are crucial for septal formation [58]. Recent literature indicates that septal defects account for over 25% of cardiac abnormalities associated with Mowat-Wilson syndrome, alongside other simpler defects such as patent ductus arteriosus [59].

Noonan syndrome (NS) is a genetic disorder caused by mutations in several genes, most commonly PTPN11, RAF1, SOS1, and BRAF. Other genes, such as RIT1, SOS2, and LZTR1, have also been identified as contributors. CHD is prevalent in NS, with diverse phenotypic manifestations depending on the affected gene. Mutations in PTPN11 are often associated with pulmonary valve stenosis and ASD, whereas RAF1 mutations are linked to hypertrophic cardiomyopathy and ASD. Mutations in SOS1 frequently result in pulmonary valve stenosis, while BRAF mutations can cause both HCM and valve abnormalities. RIT1, SOS2, and LZTR1 have been associated with unique cardiac phenotypes, including valve malformations and dilated cardiomyopathy [60].”

  1. Mital, S.; Musunuru, K.; Garg, V.; Russell, M. W.; Lanfear, D. E.; Gupta, R. M.; Hickey, K. T.; Ackerman, M. J.; Perez, M. V.; Roden, D. M.; et al. Enhancing literacy in cardiovascular genetics: a scientific statement from the American Heart Association. Cardiovasc. Genet. 2016, 9, 448–467.
  2. Heinke, D.; Isenburg, J. L.; Stallings, E. B.; Short, T. D.; Le, M.; Fisher, S.; Shan, X.; Kirby, R. S.; Nguyen, H. H.; Nestoridi, E.; et al. Prevalence of structural birth defects among infants with Down syndrome, 2013-2017: A US population-based study. Birth Defects Res. 2021, 113, 189-202.
  3. Atli, E. I.; Atli, E.; Yalcintepe, S.; Demir, S.; Kalkan, R.; Akurut, C.; Ozen, Y.; Gurkan, H. Investigation of Genetic Alterations in Congenital Heart Diseases in Prenatal Period.  Med. Genet. 2021, 9, 29-33.
  4. Duarte, V.E; Singh, M.N. Genetic syndromes associated with congenital heart disease. Heart 2024, 110, 1231-1237.
  5. Zhang, H.; Liu, L.; Tian, J. Molecular mechanisms of congenital heart disease in down syndrome. Genes Dis. 2019, 6, 372-377.
  6. Bruns, D. A.; Martinez, A. An analysis of cardiac defects and surgical interventions in 84 cases with full trisomy 18.  J. Med. Genet. A. 2016, 170A, 337-343.
  7. Møller Nielsen, A. K.; Dehn, A. M.; Hjortdal, V.; Larsen, L. A. TBX5 variants and cardiac phenotype: A systematic review of the literature and a novel variant.  J. Med. Genet. 2024, 68, 104920.
  8. Lang, Y.; Zheng, Y.; Qi, B.; Zheng, W.; Zhao, C.; Zhai, H.; Wang, G.; Luo, Z.; Li, T. Case report: Novel TBX5-related pathogenic mechanism of Holt-Oram syndrome.  Genet. 2023, 14, 1063202.
  9. St Peter, C.; Hossain, W. A.; Lovell, S.; Rafi, S. K.; Butler, M. G. Mowat-Wilson Syndrome: Case Report and Review of ZEB2Gene Variant Types, Protein Defects and Molecular Interactions.  J. Mol. Sci. 2024, 25, 2838.
  10. Ivanovski, I.; Djuric, O.; Caraffi, S. G.; Santodirocco, D.; Pollazzon, M.; Rosato, S.; Cordelli, D. M.; Abdalla, E.; Accorsi, P.; Adam, M. P.; et al. Phenotype and genotype of 87 patients with Mowat-Wilson syndrome and recommendations for care.  Med. 2018, 20, 965-975.
  11. Sun, L.; Xie, Y. M.; Wang, S. S.; Zhang, Z. W. Cardiovascular Abnormalities and Gene Mutations in Children With Noonan Syndrome.  Genet. 2022, 13, 915129.

  1. Whereas IRX4 and PITX2C also play an important role in the development of the cardiac septum, their mutations have been associated with cardiac septal defects. The authors would ideally need to comment on these factors as well.

Replay:

Acknowledging the importance of these factors, comments about PITX2C and IRX4 and their association with cardiac septal defects have been added to sections 5.11 and 5.12, respectively. This expansion offers a more detailed explanation of their roles and the implications of their mutations. Lines 519-529 for PITX2C and lines 531-538 for IRX4.

5.12. PITX2C

The PITX2 gene plays a crucial role in the development of outflow tract and atrioventricular septum. PITX2 orchestrates the specification and migration of cardiac progenitor cells, facilitating the epithelial-to-mesenchymal transition required for septal formation. Furthermore, it regulates signaling pathways such as BMP and FGF, which are essential for cushion remodeling and myocardial patterning, mutations are frequently associated with significantly diminished transcriptional activity. PITX2 abnormalities exhibit various septal defects, including unseptated atria, malformed endocardial cushions, and truncated outflow tracts, hindering normal cardiac development, also, it was identified as a causative gene for the human Axenfeld-Rieger's syndrome which also classically presents with concomitant CHD [83,84].

5.13. IRX4

The homeobox transcription factor IRX4 is a pivotal early marker of ventricular myocardium differentiation. In both the FHF and SHF of the cardiac crescent, IRX4 expression promotes the specification of left ventricular cardiomyocytes, upregulating HAND1 and downregulating ANF and alpha skeletal actin [28]. Under the regulatory influence of NKX2.5, IRX4 generates ventricular progenitor cells with the potential to differentiate into endothelial cells, smooth muscle cells, and cardiomyocytes [85]. Mutations in IRX4 can disrupt the expression of genes essential for interventricular septum formation [86].

28. Nelson DO, Jin DX, Downs KM, Kamp TJ, Lyons GE. Irx4 identifies a chamber-specific cell population that contributes to ventricular myocardium development. Dyn. 2014, 243, 381-392.

83. Ma, H.Y; Xu, J; Eng, D; Gross, M.K; Kioussi, C. Pitx2-mediated cardiac outflow tract remodeling.  Dyn. 2013, 242, 456-468.

84. Zhao, C. M.; Peng, L. Y.; Li, L.; Liu, X. Y.; Wang, J.; Zhang, X. L.; Yuan, F.; Li, R. G.; Qiu, X. B.; Yang, Y. Q. PITX2 Loss-of-Function Mutation Contributes to Congenital Endocardial Cushion Defect and Axenfeld-Rieger Syndrome. PLoS One 2015, 10, e0124409.

85. Nelson, D. O.; Lalit, P. A.; Biermann, M.; Markandeya, Y. S.; Capes, D. L.; Addesso, L.; Patel, G.; Han, T.; John, M. C.; Powers, P.A.; et al. Irx4 Marks a Multipotent, Ventricular-Specific Progenitor Cell. Stem Cells 2016, 34, 2875-2888.

86. Cheng, Z.; Wang, J.; Su, D.; Pan, H.; Huang, G.; Li, X.; Li, Z.; Shen, A.; Xie, X.; Wang, B.; et al. Two novel mutations of the IRX4 gene in patients with congenital heart disease.  Genet. 2011, 130, 657-662.

Round 2

Reviewer 1 Report

Comments and Suggestions for Authors

 The authors have addressed the raised comments and questions satisfactorily, significantly enhancing the understanding of the manuscript's initial limitations and inquiries.

Author Response

REVIEWER 1.

We thank the reviewer for the time invested in reviewing our manuscript. All the observations made were very important to improve it. Thanks again.

Reviewer 2 Report

Comments and Suggestions for Authors

The manuscript is a significant improvement over the previous version in terms of writing and data arrangement. The authors have responded appropriately to most of the issues raised by the reviewers. 

  1. 1. In Figures 1 and 2, it would be best to label the major parts of the heart structure. For example, the left and right ventricles, the left and right atria, the atrial septum, the ventricular septum, and the different valves will be easier to read. 

  1. 2. In Figure 4, the authors pointed out that BMP10 is involved in the formation of the atrial septum; are there references that confirm this? From the published references, it appears that BMP10 plays an important role in ventricular formation.

Comments on the Quality of English Language

na

Author Response

REVIEWER 2.

We thank the reviewer for the time invested in reviewing our manuscript. The observations made were very important to improve the revised version.

  1. In Figures 1 and 2, it would be best to label the major parts of the heart structure. For example, the left and right ventricles, the left and right atria, the atrial septum, the ventricular septum, and the different valves will be easier to read. 

Replay

The requested changes have been made to the images of the healthy heart; major structures have been labeled. Line 114 for ASD and line 155 for VSD.

(a) Normal heart

(b) Atrial septal defects

Figure 1. Atrial septal defect. A schematic representation of (a) normal heart; (b) atrial septal

(a) Normal heart

(b) Ventricular septal defect

Figure 2. Ventricular septal defect. A schematic representation of (a) normal heart; (b) ventricular septal defect.

  1. In Figure 4, the authors pointed out that BMP10 is involved in the formation of the atrial septum; are there references that confirm this? From the published references, it appears that BMP10 plays an important role in ventricular formation.

Replay

As noted by the reviewer, BMP-10 expression is indeed localized to the trabeculated region of the common ventricular chamber. There was a confusion regarding nomenclature, as BMP family members. Thanks for your observation, we have corrected this point in Figure 4 (lines 202- 207) and added one necessary reference.

Reference:

37. Neuhaus, H.; Rosen, V.; Thies, R. S. Heart specific expression of mouse BMP-10 a novel member of the TGF-be
